# Nuclear soluble cGAS senses double-stranded DNA virus infection

Yakun Wu[1,2], Kun Song[1,2], Wenzhuo Hao[1], Jack Li[1], Lingyan Wang[1] & Shitao Li [1✉]

The DNA sensor cGAS detects cytosolic DNA and instigates type I interferon (IFN) expression. Recent studies find that cGAS also localizes in the nucleus and binds the chromatin. Despite the mechanism controlling nuclear cGAS activation is well elucidated, whether nuclear cGAS participates in DNA sensing is unclear. Here, we report that herpes simplex virus 1 (HSV-1) infection caused the release of cGAS from the chromatin into the nuclear soluble fraction. Like its cytosolic counterpart, the leaked nuclear soluble cGAS also sensed viral DNA, produced cGAMP, and induced mRNA expression of type I IFN and interferon-stimulated genes. Consistently, the nuclear soluble cGAS limited HSV-1 infection. Furthermore, enzyme-deficient mutation (D307A) or cGAS inhibitor RU.251 abolished nuclear cGAS-mediated innate immune responses, suggesting that enzymatic activity is also required for nuclear soluble cGAS. Taken all together, our study demonstrates that nuclear soluble cGAS acts as a nuclear DNA sensor detecting nuclear-replicating DNA viruses.

[1] Department of Microbiology and Immunology, Tulane University, New Orleans, LA 70112, USA. [2] These authors contributed equally: Yakun Wu, Kun Song.
✉email: sli38@tulane.edu

ytosolic pathogenic DNA or self-DNA from cellular damage is detected by the DNA sensor, cyclic GMP-AMP synthase (cGAS)[1]. cGAS produces the second messenger cyclic GMP-AMP (cGAMP), which binds to the endoplasmic reticulum membrane protein, stimulator of interferon genes (STING)[2–5]. Subsequently, STING dimerizes and traffics from the ER to the Golgi. At the Golgi, STING activates downstream TANK-binding kinase 1 (TBK1) and the IκB kinase (IKK) complex. The activated TBK1 and the IKK triggers the dimerization and nuclear translocation of interferon regulatory factors (IRFs) and NF-κB transcriptional factors, respectively. In the nucleus, IRFs and NF-κB co-opt to activate type I IFN gene expression[6–10].

Excessive host DNA can activate the cGAS signaling pathway, leading to aberrant IFN activation and autoimmune diseases, such as Aicardi-Goutieres syndrome (AGS)[11]. Therefore, cells must render cGAS inert to host genomic DNA, and paradoxically, at the same time, cells need to keep cGAS agile to foreign DNA. The old paradigm is that host DNA is normally restricted to cellular compartments, such as the nucleus and the mitochondria. As cGAS was first thought to be a solely cytosolic protein, the physical barrier is the explanation for the inaccessibility of cGAS to host DNA[12–14]. However, recent studies found other subcellular localizations of cGAS, including predominantly in the nucleus[15], on the plasma membrane[16], mitosis-associated nuclear localization[17,18], or phosphorylation-mediated cytosolic retention[19]. Nonetheless, the nuclear localization of cGAS has been validated by many independent laboratories[15,18,20], which is of particular interest as the nucleus is a DNA-rich environment. Nuclear cGAS is bound to the chromatin in the nucleus by binding to the H2A-H2B dimer of the nucleosome, which immobilizes cGAS on the chromatin; thus, cGAS cannot access the nearby DNA to form an active dimer[21–26]. In addition, the phosphorylation of the N-terminus of cGAS and the S291 site at the C-terminus further prevents cGAS from activation during mitosis[27,28]. Although the mechanism of nuclear cGAS inhibition is well elucidated, the role of nuclear cGAS in DNA sensing, especially in the settings of infection, is unknown.

Here, we found that endogenous cGAS was tethered to the chromatin in multiple cell lines and HSV-1 infection caused the release of cGAS from the chromatin into the nuclear soluble fraction. The nuclear soluble cGAS bound viral DNA and produced cGAMP. Furthermore, cells exclusively expressing nuclear cGAS responded to HSV-1 infection and activated type I IFN expression. Collectively, our study suggests that HSV-1 infection leads to cGAS release from the chromatin tethering; in turn, the nuclear soluble cGAS senses viral DNA and activates type I IFN to suppress viral infection. Our study reveals the role of nuclear cGAS in host defense to DNA virus infection.

## Results

### Endogenous cGAS localizes in the cytoplasm and the nucleus.
To determine the size of endogenous cGAS protein complex, we performed a sucrose gradient ultracentrifugation for cell lysates of RAW264.7 macrophages. Unexpectedly, the majority of cGAS proteins were distributed in the high molecular weight fractions with a high sedimentation rate (Fig. 1a). Interestingly, histone H3 co-fractionated with cGAS in most fractions with high molecular weight (Fig. 1a), suggesting that cGAS might associate with histones or the chromatin[15,27,29]. To further determine the subcellular localization of endogenous cGAS, we fractionated the cell lysates into five fractions: cytosol, membrane, nuclear soluble, chromatin-bound, and cytoskeletal. The nuclear soluble fraction is extracted in a low salt concentration and does not contain histones and nucleosomes, whereas the chromatin-bound fraction comprises nucleosomes with a high salt concentration extract

condition. cGAS was found in the cytosol and the nucleus of H1299 cells (Fig. 1b). Consistent with a previous report[30], cGAS localized in the chromatin-bound nuclear fraction but not the nuclear soluble fraction (Fig. 1b). Similar results were observed in THP-1 cells (Supplementary Fig. 1a). We also examined whether DNA stimulation altered cGAS distribution in each fraction. However, the amount of cGAS in each fraction was comparable in cells with vs. without DNA stimulation (Fig. 1b and Supplementary Fig. 1a).

Next, we examined endogenous cGAS localization in cells by immunofluorescence assays (IFA). We first validated a newly developed anti-human cGAS antibody (Cell Signaling Technology, #79978 S) by Western blotting (Supplementary Fig. 1b) and IFA (Fig. 1c) in cGAS wild type and knockout H1299 cells. Using this antibody, we found that cGAS was localized in the cytosol, nucleus, around nucleoli, micronuclei, chromosome, chromatin bridge, and perinuclear region (Fig. 1d). Furthermore, we examined the endogenous cGAS localization in several other cell lines. However, the patterns of endogenous cGAS distribution varied in different cell lines, irrespective of cytosolic DNA stimulation (Supplementary Fig. 1c–f). In agreement with previous studies, we found that endogenous cGAS localized in the nucleus in all tested cell lines, suggesting a potential role of nuclear cGAS.

### The N-terminus and NES regulate cGAS nuclear localization.
Previous efforts have been made to determine the region responsible for cGAS nuclear localization. One study suggested that the region of amino acids 161-212 is essential for cytosolic retention of human cGAS[18]. There is a conserved classic nuclear export signal (NES) "LxxxLxxLxL/I" within amino acids 161-212 (Supplementary Fig. 1g)[31]. To examine the role of the NES in cGAS subcellular localization, we deleted the NES in cGAS. IFA assays showed that the NES deletion led to a slight increase of nuclear cGAS (Supplementary Fig. 1h). We further mutated two leucines in the NES into arginine and lysine (LL/RK) in mouse cGAS, respectively. These mutations not only disrupted the NES but also converted the NES into a nuclear localization signal (NLS) (Fig. 1e). IFA assays showed that the LL/RK mutation dramatically increased cGAS nuclear localization (Fig. 1f). However, there were still approximately 37% of cells in which cGAS resided in both the nucleus and the cytoplasm (Fig. 1f). These data suggest that the NES might be required but not sufficient for cGAS cytosolic localization.

We further examined which domain might control cGAS nuclear localization (Supplementary Fig. 1g). We stably transfected the N-terminal domain (N) and the N deletion mutant (delN) of cGAS into HEK293 cells. Subcellular fractionation found that the N and delN showed distinct localizations (Supplementary Fig. 1i). The N was mainly distributed in the cytosol with a small portion in the nuclear soluble fraction. As reported recently, the delN localized in the membrane fraction due to the exposure of mitochondrial targeting signal (MTS) (Supplementary Fig. 1g). Furthermore, delN was also found in the chromatin-bound and cytoskeletal fractions. Thus, we mutated the two leucines in the NES into arginine and lysine in delN of cGAS (delN-LL/RK) to disrupt the NES and MTS. Interestingly, subcellular fractionation found that the delN-LL/RK is exclusively expressed in the chromatin-bound fraction (Fig. 1g). Like endogenous cGAS, cytosolic DNA stimulation had little effect on the localization of delN-LL/RK (Fig. 1g). Overall, our data suggest that both NES and the N-terminal domain regulate cGAS subcellular localization.

### HSV-1 infection causes cGAS release from the chromatin to the nuclear soluble fraction.
As most DNA viruses are uncoated

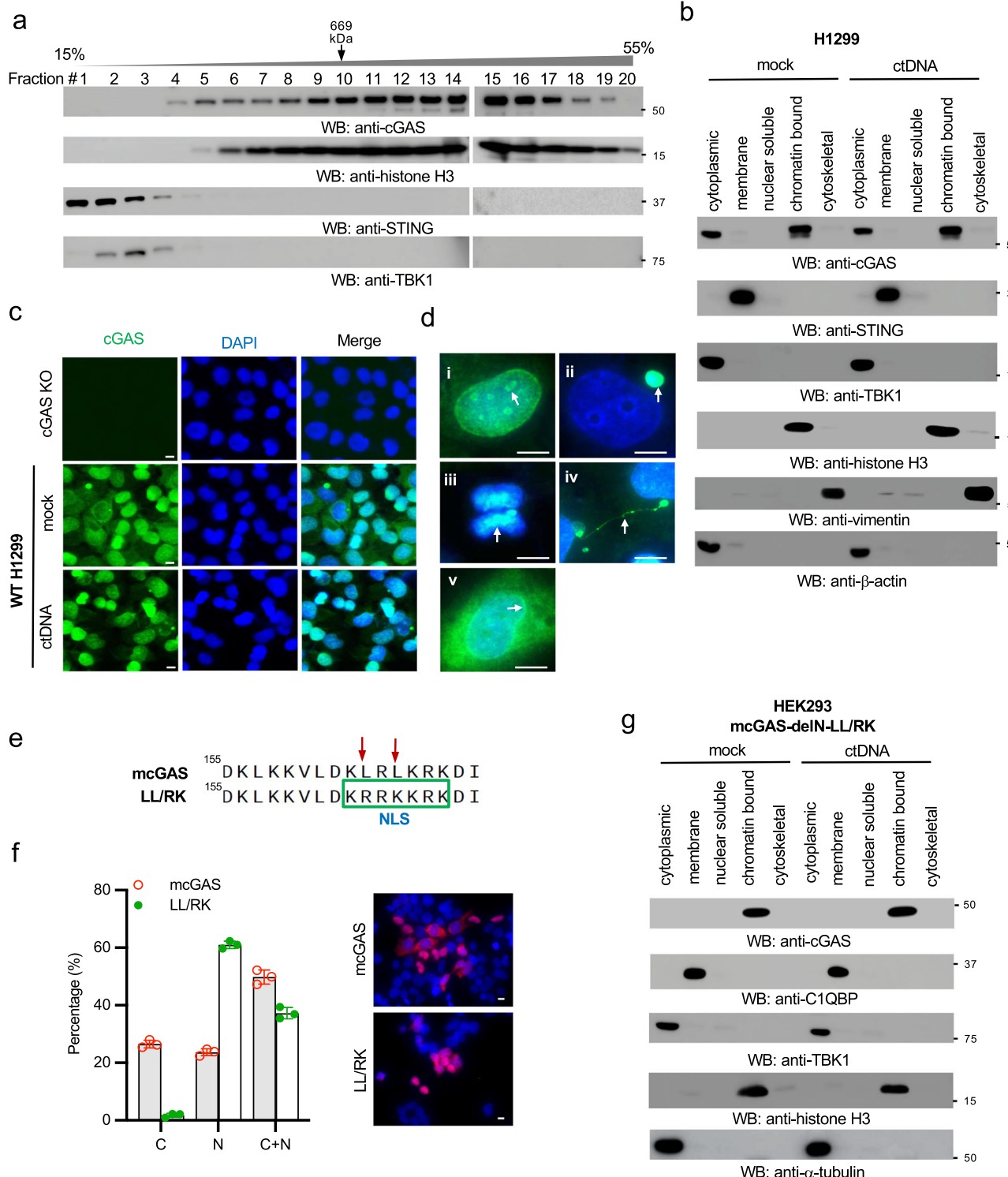

and then replicate in the nucleus, it would be an opportunity for the host to detect viral DNA in the nucleus during viral infection. However, nuclear cGAS is immobilized on the chromatin[21–25]. We hypothesized that DNA virus infection in the nucleus might cause cGAS release from the chromatin. In this regard, we infected RAW 264.7 cells with herpes simplex virus type 1 (HSV-1), a DNA virus that transcribes and replicates in the nucleus. Infection with two HSV-1 strains, McKrae (Fig. 2a) and KOS (Supplementary Fig. 2a) caused a portion of cGAS to translocate to the nuclear soluble fraction. By contrast, cytosolic DNA stimulation by transfection had little effect on cGAS subcellular

localization in RAW264.7 macrophages (Supplementary Fig. 2b). Furthermore, HSV-1 infection also led to accumulation of nuclear soluble cGAS in THP-1 cells and mouse bone marrow-derived macrophages (Supplementary Fig. 2c, d).

Next, we examined the effects of two DNA viruses, adenovirus (AdV) and vaccinia virus (VACV), and two RNA viruses, influenza A viruses (IAV) and vesicular stomatitis virus (VSV), on cGAS subcellular localizations. AdV and IAV replicate in the nucleus while VACV and VSV replicate in the cytoplasm. Like HSV-1, AdV infection induced nuclear soluble cGAS in RAW264.7 macrophages (Fig. 2b). However, VACV, IAV, and

**Fig. 1 The N-terminal domain and NES regulate cGAS nuclear localization. a** The cell lysates of RAW264.7 macrophages were separated by 15–55% sucrose density centrifugation. Fractions were blotted as indicated. The fraction of thyroglobulin (669 kDa), a protein standard, was indicated. **b** H1299 cells were stimulated with or without 1 μg/mL ctDNA by transfection for 4 h. Then, the cell lysates were fractionated into five fractions: cytoplasmic, membrane, nuclear soluble, chromatin-bound, and cytoskeletal. The fractions were blotted as indicated. STING: membrane marker; TBK1 and β-actin: cytosolic marker; H3: nuclear marker; vimentin: cytoskeletal marker. **c** Wild type (WT) and cGAS knockout (KO) H1299 cells were either mock stimulated or transfected with ctDNA. After 4 h, cells were fixed and stained as indicated. cGAS: green; DAPI, blue. Bar = 10 μm. **d** Representative cGAS localization in unstimulated H1299 WT cells in (**c**). (**i**) surrounding nucleoli; (**ii**) micronucleus; (**iii**) chromosome; (**iv**) chromatin bridge; (**v**) perinuclear region. Arrows indicate each distinct localization in (**i**) to (**v**). cGAS: green; DAPI, blue. Bar = 10 μm. **e** Schematic of the LL/RK mutation in the nuclear export signal of mouse cGAS (mcGAS). Red arrows indicate the mutated sites, and the green frame indicates the introduction of a NLS caused by the mutation. **f** IFA of HEK293 cells stably expressing FLAG-tagged mcGAS or the indicated LL/RK mutant. FLAG: red; DAPI, blue. Bar = 10 μm. The summary of the subcellular localization of mcGAS and the LL/RK mutant was shown in the left panel. C: cytosolic; N: nuclear; C + N: cytosolic and nuclear. Data represent means ± s.d. of three independent experiments (> 200 cells were counted in each field and five fields were counted per experiment). **g** HEK293 cells stably expressing the delN with LL/RK mutation of mcGAS (mcGAS-delN-LL/RK) were transfected with or without 1 μg/mL ctDNA for 4 h. Then, the cell lysates were fractionated into five fractions: cytoplasmic, membrane, nuclear soluble, chromatin-bound, and cytoskeletal. The fractions were blotted as indicated. C1QBP: membrane marker; TBK1 and α-tubulin: cytosolic marker; H3: nuclear marker.

VSV failed to induce cGAS translocation to the nuclear soluble fraction in RAW264.7 macrophages (Fig. 2c–e), suggesting that DNA virus replication in the nucleus is required for cGAS release from the chromatin to the nuclear soluble fraction.

**Inhibiting HSV-1 replication blocks cGAS release from the chromatin**. To determine the mechanism by which HSV-1 induces cGAS release from the chromatin, we first examined the effects of HSV-1 KOS d109 mutant virus in which the IFN-suppression viral genes are deleted[32]. As shown in Fig. 3a, the d109 mutant induced nuclear soluble cGAS, suggesting these IFN-suppression viral proteins have little effect on cGAS localization. Next, we examined the effects of viral replication on cGAS release. The HSV-1 DNA polymerase inhibitor acyclovir was used to inhibit viral replication. As predicted, viral protein ICP8 expression reduced after acyclovir treatment. More interestingly, acyclovir blocked viral infection-induced nuclear soluble cGAS (Fig. 3b).

As HSV-1 infection can cause nuclear stress and host DNA damage, we suspected that DNA damage might induce the release of cGAS from chromatin tethering. In this regard, we treated RAW264.7 macrophages with a DNA damage agent, cisplatin[33]. Cisplatin induced γH2AX (phosphorylated S139 H2AX histone), a hallmark of DNA damage; however, cisplatin failed to induce nuclear soluble cGAS (Fig. 3c). To exclude the effects of cell viability on cGAS subcellular localization, we performed MTT assays in cells treated with cisplatin or HSV-1. As expected, cisplatin reduced 90% cell viability 24 h after treatment (Fig. 3d). However, HSV-1 infection had little effect on cell viability (Fig. 3d). Taken together, these data suggest that DNA virus nuclear replication, but not IFN-suppression viral proteins and DNA damage, causes cGAS release from the chromatin.

**Nuclear soluble cGAS is constitutively active**. It has been reported that R241E mutation results in the release of cGAS from the chromatin[15]. We stably expressed cGAS R241E mutant in HEK293 cells that are lacking endogenous cGAS. As reported previously[15], the R241E mutant was only present in the cytosol and the nuclear soluble fractions (Supplementary Fig. 3a) and produced a significant amount of cGAMP without DNA ligand stimulation (Supplementary Fig. 3b), suggesting the nuclear soluble cGAS is constitutively active.

Next, we examined whether the nuclear soluble cGAS induced by HSV-1 was active. We transfected RAW264.7 cells with ctDNA or infected the cells with HSV-1 followed by subcellular fractionation. Then, we performed in vitro cGAS enzymatic assays to assess the activity of cGAS in each fraction. The cytosolic cGAS

had comparable enzymatic activities in cells infected with HSV-1 and transfected with ctDNA (Supplementary Fig. 3c). However, a significantly higher level of cGAMP was generated by cGAS in the nuclear soluble extract of cells infected with HSV-1 (Supplementary Fig. 3c), suggesting the active state of the nuclear soluble cGAS. Furthermore, we compared the cGAS activity in the nuclear soluble fraction in cells infected with different viruses. As shown in Supplementary Fig. 3d, cGAMP production was only detected in the nuclear soluble extract from cells infected with HSV-1, but not other tested viruses. These data are consistent with the subcellular fraction results that IAV, VSV, and VACV failed to induce nuclear soluble cGAS.

**Establish a cell line exclusively expressing nuclear cGAS**. To examine the role of nuclear cGAS, we generated a cell line exclusively expressing nuclear cGAS to exclude the interference by cytosolic cGAS. In this regard, we fused the NLS of SV40 large T to the C-terminus of cGAS (cGAS-NLS). We then stably transfected these constructs into HEK293 cells. Although most cGAS-NLS proteins resided in the nuclear fraction, there was still a fair amount of cGAS in the cytosol (Fig. 4a). Thus, to further block cytosolic localization of cGAS, we fused the NLS to the LL/RK mutant (LL/RK-NLS). As shown in Fig. 4a, the LL/RK-NLS was exclusively in the nuclear fraction. To further exclude any newly synthesized cytosolic cGAS, we generated an inducible system for cGAS and the LL/RK-NLS (Fig. 4b) and stably transfected them into HEK293 cells (Supplementary Fig. 4a). As shown in Fig. 4c, cGAS protein degraded after 24 h, but was stable within 8 h after doxycycline removal. Furthermore, immunofluorescence assays showed that only the LL/RK-NLS proteins were exclusively localized in the nucleus (Supplementary Fig. 4b).

To determine the functional role of nuclear soluble cGAS, we applied the inducible cGAS expression system into RAW264.7 knockout macrophages. We first generated cGAS knockout in RAW264.7 macrophages (Supplementary Fig. 4c). cGAS knockout cells failed to produce IFNβ and ISGs, such as IP-10 and RANTES, when infected with HSV-1 (Supplementary Fig. 4d). Next, we reconstituted the inducible cGAS and the LL/RK-NLS mutant in cGAS knockout RAW264.7 macrophages (hereinafter referred to as KO(cGAS) and KO(LL/RK-NLS), respectively) (Supplementary Fig. 4e). Subcellular fractionation showed that LL/RK-NLS, but not the wild type cGAS, only resided in the chromatin-bound extract of the reconstituted cells after Doxycycline induction (Fig. 4d).

**Nuclear soluble cGAS senses HSV-1 and activates innate immune responses**. To examine the role of nuclear cGAS in

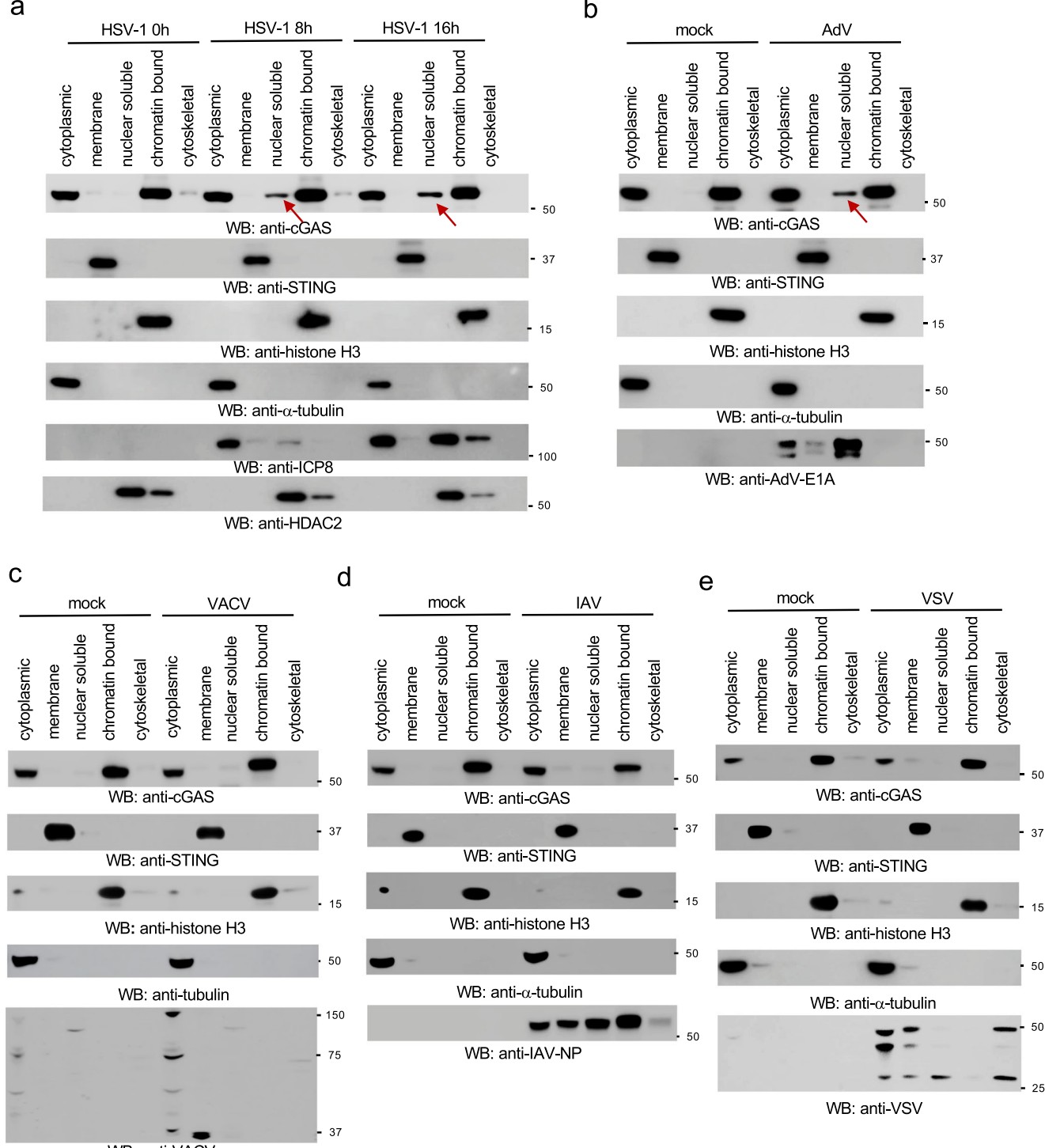

**Fig. 2 HSV-1 and AdV cause cGAS release from the chromatin to the nuclear soluble fraction. a** RAW 264.7 macrophages were infected with 1 MOI of HSV-1 McRae for designated times, and then the subcellular fractions of cell lysates were blotted as indicated. The arrow indicates nuclear soluble cGAS after viral infection. STING: membrane marker; α-tubulin: cytosolic marker; H3: nuclear marker; ICP8: HSV-1 viral protein; HDAC2: nuclear marker. **b**–**e** RAW 264.7 macrophages were infected with 1 MOI of AdV (**b**), VACV (**c**), IAV (**d**) or VSV (**e**) for 16 h. The subcellular fractions were blotted as indicated. The arrow indicates nuclear soluble cGAS after viral infection.

HSV-1 infection, we infected the cGAS KO(LL/RK-NLS) RAW 264.7 cells with HSV-1. HSV-1 infection induced cGAS release from the chromatin to the nuclear soluble fraction (Fig. 5a), which further corroborates that the nuclear soluble cGAS leaks from the chromatin. Next, we compared innate immune responses to HSV-1 infection in the cGAS KO, KO(cGAS), and

KO(LL/RK-NLS) RAW264.7 cells. We infected these cells with HSV-1 KOS d109 because this mutant virus induces much stronger innate immune responses due to the deletion of the IFN-suppressing viral genes[32]. As expected, cGAS KO cells failed to respond to HSV-1 infection; however, cGAS and LL/RK-NLS restored the mRNA expression of IFNβ, RANTES, and IP-10

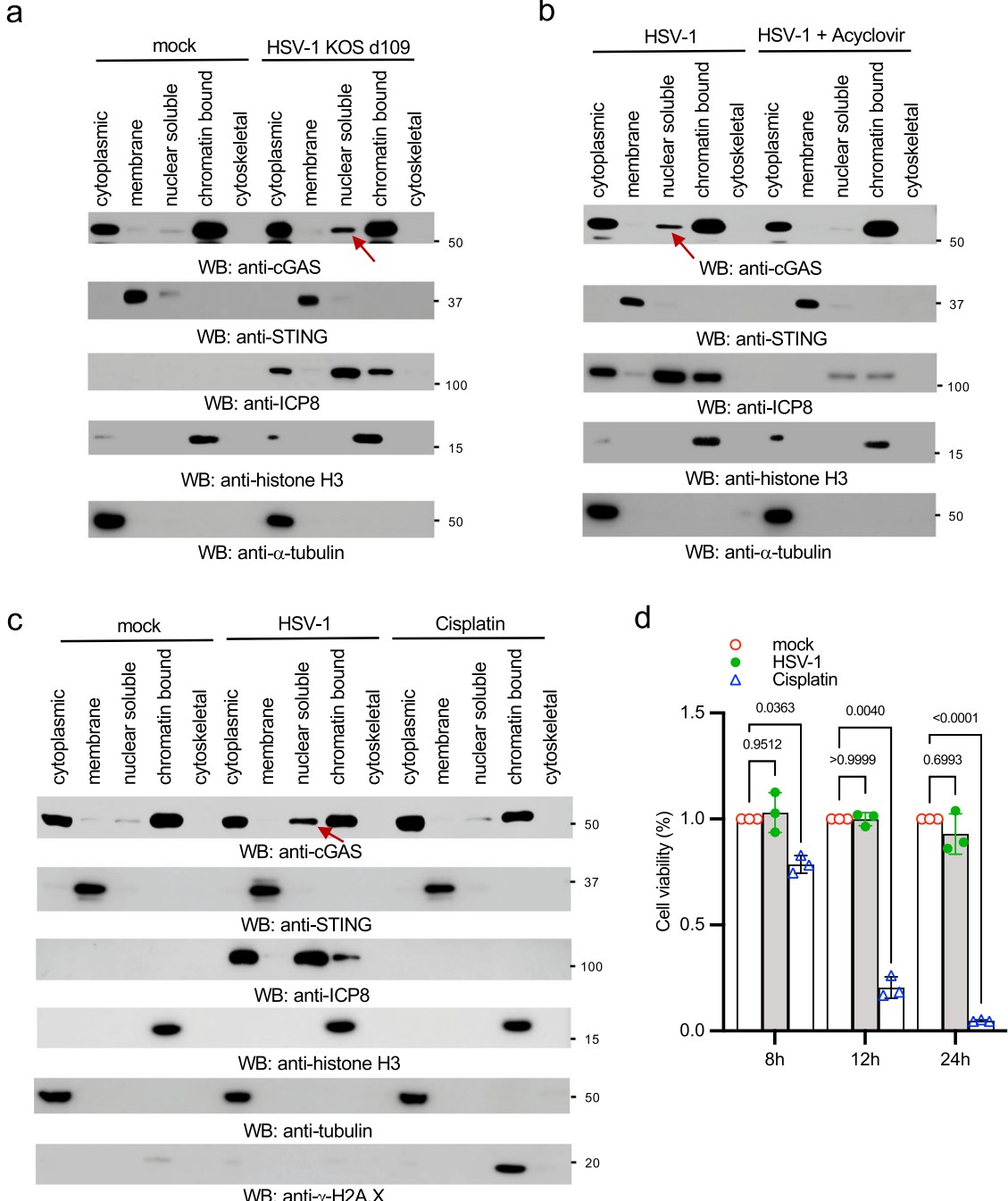

**Fig. 3 Inhibition of HSV-1 replication blocks cGAS release from the chromatin. a** RAW 264.7 cells were mock-infected or infected with 1 MOI of HSV-1 KOS d109 for 16 h. Cell lysates were fractionated and blotted as indicated. **b** RAW 264.7 cells were pretreated without or with 8 μg/mL of acyclovir for 16 h, followed by infection with 1 MOI of HSV-1 McKrae for 12 h. Cell lysates were fractionated and blotted as indicated. The arrow indicates nuclear soluble cGAS. **c** RAW 264.7 macrophages were treated with dimethylformamide (DMF) as a vehicle mock control, 50 μM cisplatin for 4 h, or infected with HSV-1 McKrae for 12 h. Cell lysates were fractionated and blotted as indicated. The arrow indicates nuclear soluble cGAS. STING: membrane marker; α-tubulin: cytosolic marker; H3: nuclear marker; ICP8: HSV-1 viral protein; γ-H2A.X: DNA damage marker. The arrow indicates nuclear soluble cGAS. **d** RAW 264.7 macrophages were treated with DMF (mock control), 50 μM cisplatin or infected with HSV-1 McKrae for indicated times. Then, cell viability was determined by MTT assays. Data represent means ± s.d. of three independent experiments. The *P*-value was calculated by two-way ANOVA followed by Sidak's multiple comparisons test.

(Fig. 5b, c and Supplementary Fig. 5a). The reconstitution of cGAS and LL/RK-NLS also rescued TBK1 phosphorylation, the hallmark of activation of IFN production pathways, in cGAS knockout cells (Fig. 5d). Furthermore, KO(LL/RK-NLS) RAW264.7 cells produced a significant amount of cGAMP after HSV-1 infection (Fig. 5e). By contrast, transfected ctDNA failed

to activate innate immune response in KO(LL/RK-NLS) RAW264.7 macrophages (Supplementary Fig. 5b), suggesting that nuclear cGAS is activated by HSV-1 infection, but not cytosolic DNA.

Next, we examined whether cGAS enzymatic activity was required for nuclear soluble cGAS-mediated innate immune

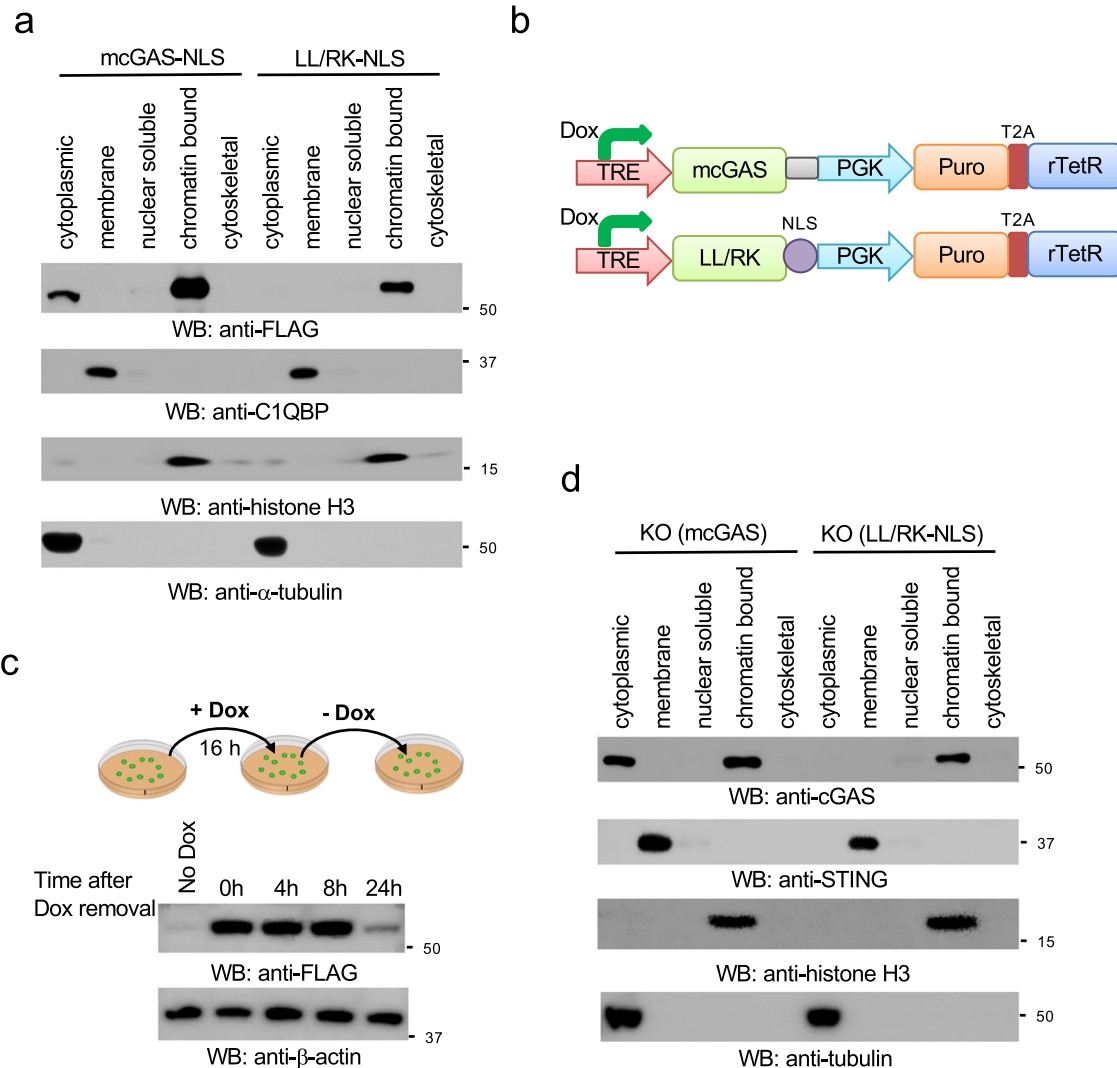

**Fig. 4 Generation of a stable cell line exclusively expressing nuclear cGAS. a** HEK293 cells stably expressing mouse cGAS fused with a C-terminal NLS (mcGAS-NLS) or the LL/RK-NLS mutant were fractionated and blotted as indicated. C1QBP: membrane marker; α-tubulin: cytosolic marker; H3: nuclear marker. **b** Schematic of the doxycycline (Dox)-induced mcGAS and LL/RK-NLS constructs. TRE: Tet Response Element; Puro: puromycin; T2A: Thosea asigna virus 2A-like peptide. **c** HEK293 cells stably expressing mcGAS were treated with 2 μg/mL Dox for 24 h. Cells were harvested at the indicated times after Dox removal. Cell lysates were blotted as indicated. **d** Stable cGAS knockout RAW264.7 cells reconstituted with the inducible mcGAS or the LL/RK-NLS mutant were treated with 2 μg/mL Dox for 24 h. Then cells were fractionated into five fractions and blotted as indicated. STING: membrane marker; α-tubulin: cytosolic marker; H3: nuclear marker.

responses. The KO(LL/RK-NLS) RAW264.7 cells were treated with DMSO or the cGAS inhibitor RU.521 followed by HSV-1 KOS d109 infection. As shown in Fig. 5f, RU.521 inhibited mRNA expression of IFNβ, RANTES, and IP-10 induced by HSV-1 KOS d109. Consistently, the enzyme-deficient mutation (D307A) of cGAS also abolished mRNA expression of IFNβ, RANTES, and IP-10 induced by HSV-1 KOS d109 in the KO(LL/RK-NLS) RAW264.7 cells (Fig. 5g). Furthermore, we examined whether the nuclear cGAS sensing required downstream STING. We stably transfected LL/RK-NLS into HEK293 and HEK293T cells (Supplementary Fig. 5c). As HEK293T cells do not express STING (Supplementary Fig. 5c), we used it as the control for HEK293 cells. After Dox induction, HEK293 cells were more resistant to HSV-1 infection than HEK293T cells, indicated by the reduced ICP8 protein levels in HEK293 cells (Supplementary Fig. 5d). Consistently, HSV-1 infection induced IFNβ mRNA expression in HEK293 cells, but not HEK293T cells (Supplementary Fig. 5e). These data suggest that nuclear soluble cGAS instigates innate immune responses via STING.

It has been reported that cGAS can bind host genomic DNA; however, whether nuclear soluble cGAS binds viral DNA is unknown. Thus, we infected the KO(LL/RK-NLS) cells with the HSV-1 carrying a GFP. After infection, the nuclear soluble extracts were subject to chromatin immunoprecipitation (ChIP) assay. ChIP assays found that cGAS bound GFP and HSV-1 VP16 DNA (Supplementary Fig. 5f), suggesting that cGAS can sense HSV-1 DNA in the nuclear soluble fraction.

**Nuclear soluble cGAS inhibits HSV-1 infection**. To examine whether nuclear soluble cGAS inhibits HSV-1 infection, cGAS KO, KO(cGAS), KO(LL/RK-NLS), and KO(LL/RK-NLS)/ D307A cells were infected with HSV-1-GFP. As shown in Fig. 6a, cGAS and the LL/RK-NLS mutant, but not LL/RK-NLS-D307A, rescued the antiviral activity in cGAS knockout cells, evidenced by the reduced number of GFP staining cells. Consistently, the reconstitution of LL/RK-NLS also reduced the expression of viral RNA (Fig. 6b), viral protein (Fig. 6c), and the production

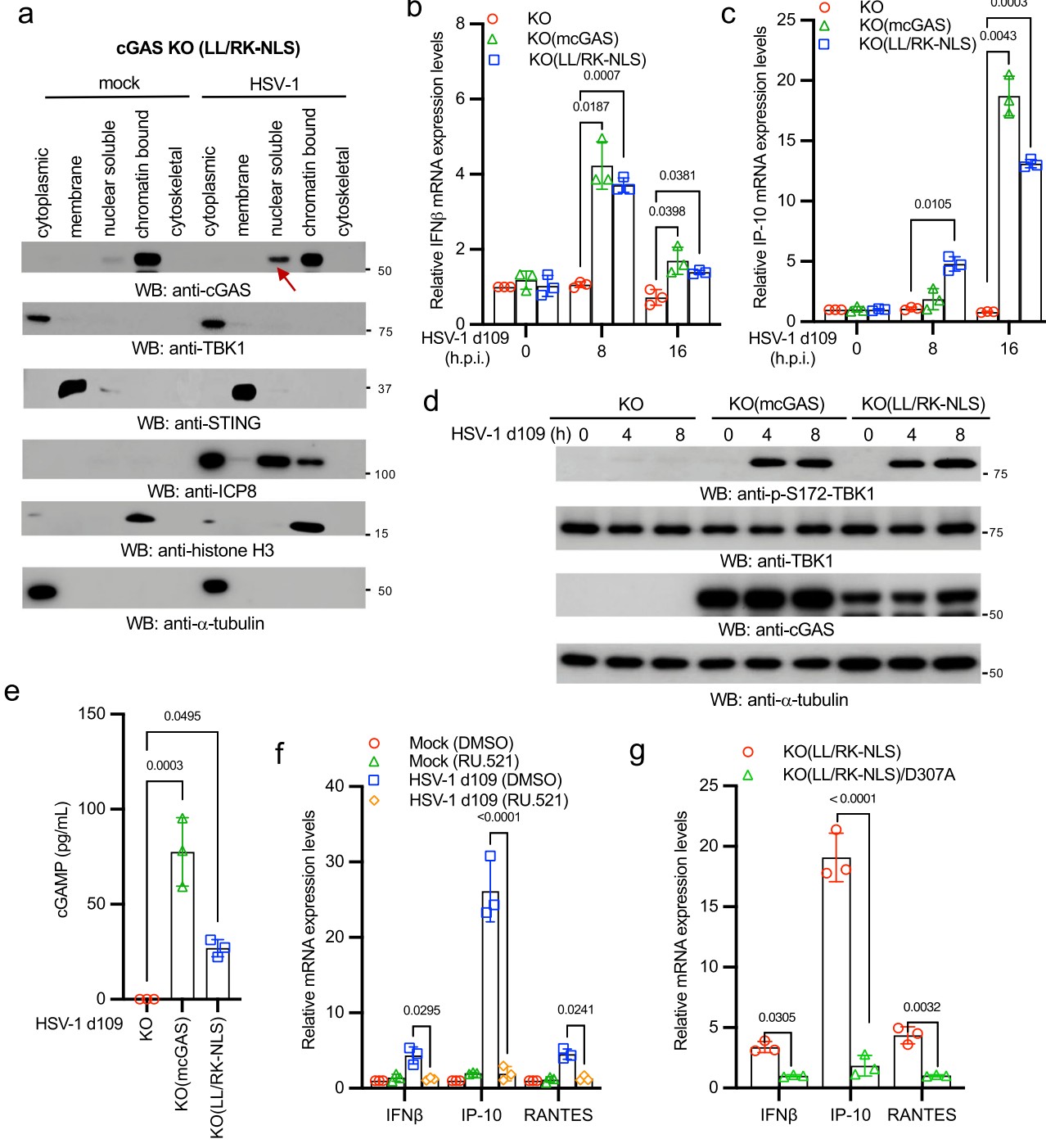

of viral particles (Fig. 6d), suggesting that nuclear soluble cGAS limits HSV-1 infection. Furthermore, we examined the role of nuclear cGAS in other viruses. We infected cGAS KO, KO(cGAS), and KO(LL/RK-NLS) cells with HSV-1, VACV, VSV, and IAV reporter viruses. The infection activity of HSV-1, but not other viruses, reduced in the KO(LL/RK-NLS) cells (Fig. 6e), which is consistent with that only HSV-1 induces nuclear soluble cGAS. By contrast, reconstitution of wild type cGAS in the knockout cells inhibited VACV and VSV infection (Fig. 6e), which is similar to the wild type cells (Supplementary Fig. 6a). Taken together, our data demonstrate that nuclear soluble cGAS can sense DNA virus infection in the nucleus, instigate innate immune response, and inhibit viral infection.

## Discussion

Several cGAS localizations have been reported, such as predominantly in the cytoplasm[1], predominantly in the nucleus[15], on the plasma membrane[16], in the cytoplasm and the nucleus[18], mitosis-associated nuclear localization[17,18], or phosphorylation-mediated cytosolic retention[19]. Although the discrepancy might be partially due to the cell type, the major reason is that most of the conclusions are based on imaging of the epitope-tagged to cGAS. We revisited the subcellular localization of endogenous cGAS by using a validated antibody. Consistent with some previous studies, we found endogenous cGAS localized in the cytoplasm, nucleus, chromosome, chromatin bridge, and micronuclei. However, different cell lines showed distinct subcellular localization patterns. For example, the localization pattern of cGAS is

**Fig. 5 Nuclear soluble cGAS senses HSV-1 infection and instigates innate immune response. a** The cGAS KO(LL/RK-NLS) RAW264.7 cells were treated with 2 μg/mL Dox for 24 h, followed by mock-infection or infection with 1 MOI of HSV-1 McKrae for 16 h. Then, cells were fractionated into five fractions and blotted as indicated. STING: membrane marker; TBK1 and α-tubulin: cytosolic marker; H3: nuclear marker. Red arrow indicates nuclear soluble cGAS. **b**, **c** The cGAS KO, KO(mcGAS), KO(LL/RK-NLS) RAW264.7 cells were treated with Dox for 24 h and then infected with 1 MOI of HSV-1 KOS d109 for designated times. Real-time PCR was performed to determine the relative mRNA levels of IFNβ (**b**) and IP-10 (**c**). Data represent means ± s.d. of three independent experiments. The P-value was calculated by two-way ANOVA followed by Dunnett's multiple comparisons test. **d** The cGAS KO, KO(mcGAS), KO(LL/RK-NLS) RAW264.7 macrophages were treated with Dox for 24 h and then infected with 1 MOI of HSV-1 KOS d109 for indicated times. Cell lysates were collected and blotted as indicated. **e** cGAS KO, KO(mcGAS), KO(LL/RK-NLS) RAW264.7 macrophages were treated with Dox for 24 h and then infected with 1 MOI of HSV-1 KOS d109 for 8 h. The amount of cGAMP in each cell line was determined by ELISA assays. All experiments were repeated three times. The P value was calculated by one-way ANOVA followed by Tukey's multiple comparisons test. **f** The cGAS KO(LL/RK-NLS) RAW264.7 cells were treated with DMSO or 25 μM RU.521 for 16 h. Then, cells were mock-infected or infected with HSV-1 KOS d109 for 16 h. Real-time PCR was performed to determine the relative mRNA levels of IFNβ, IP-10, and RANTES. Data represent means ± s.d. of three independent experiments. The P-value was calculated by two-way ANOVA followed by Tukey's multiple comparisons test. **g** The cGAS KO(LL/RK-NLS) and KO(LL/RK-NLS)/D307A cells were treated with Dox for 24 h and then mock-infected or infected with HSV-1 KOS d109 for 16 h. Real-time PCR was performed to determine the relative mRNA levels of IFNβ, IP-10, and RANTES. Data represent means ± s.d. of three independent experiments. The P-value was calculated by two-way ANOVA followed by Sidak's multiple comparisons test.

consistent in most HFF-1 cells but cGAS could be either cytosolic or nuclear in H1299 cells. Our data imply additional mechanisms might be required to regulate endogenous cGAS localization in different types of cells.

Currently, it is not clear how cGAS shuttles between the cytosol and the nucleus. Several studies proposed that cGAS nuclear localization results from nuclear envelop breakdown in mitosis or nuclear envelope rupture in interphase[34,35]. A recent study reported that the export of nuclear cGAS to the cytosol is required for cytosolic DNA sensing based on the observation of accumulation of cytosolic cGAS after DNA stimulation[31]. Although the mutation of the NES moderately altered cGAS subcellular localizations, we did not observe an accumulation of cytosolic cGAS in multiple cell lines after DNA stimulation. However, our approaches cannot exclude the nucleocytoplasmic shuttling of cGAS. Nonetheless, there is a fair amount of cytosolic cGAS present in the cytosol of unstimulated cells. Another recent study showed that the collided ribosomes induced nuclear cGAS translocation to the cytosol under translation stress[36]. The critical question for these models is how cytosolic DNA or translation stress transduces a signal to nuclear cGAS proteins which are chromatin-bound. Logically, chromatin-bound cGAS would be first released into the nuclear soluble fraction, like the R222E and R241E mutants. Then, the nuclear soluble cGAS is exported into the cytosol. However, nuclear soluble cGAS is barely seen during cytosolic DNA stimulation. Whether and how cytosolic DNA stimulation induces cGAS nuclear export needs further investigation in the future.

It has been reported that nuclear cGAS is involved in DNA homologous recombination and DNA replication[19,37]. However, the function of nuclear cGAS in host defense and viral infection is unknown. Recent studies showed that the murine R222 (R236 in human cGAS) and murine R241 (R255 in human) sites are the critical sites for the binding to the H2A-H2B dimer[15,21–25]. The mutation of these two arginines to glutamic acids leads to the release of chromatin-bound cGAS into the nuclear soluble fraction and constitutive cGAS activation[15], suggesting that nuclear cGAS can be activated. However, whether nuclear cGAS can be activated by virus is unknown. Our data now demonstrate that HSV-1 infection induces cGAS release from the chromatin. The "free" nuclear cGAS is in a DNA-rich environment, and the untethering is sufficient to activate cGAS by either host or viral DNA (Fig. 6f). Our study also indicates that nuclear soluble cGAS-mediated innate immune response is STING dependent. Several recent studies reported the inner nuclear membrane localization of STING[38,39], which raises an interesting question of whether nuclear STING is a receptor for nuclear cGAMP. Although it has been suggested that cGAMP might diffuse freely

through nuclear pores[18], we cannot exclude the possibility that nuclear cGAMP binds and activates nuclear STING. However, it is technically difficult to exclude the interference of ER-resident STING because of the continuous membrane system between the nuclear and ER membranes. Future studies are warranted to investigate the role of nuclear STING.

Our model suggests that the nuclear cGAS senses the viruses or other invading pathogens that can evade the surveillance of cytosolic cGAS by uncoating in the nucleus. Indeed, most DNA viruses, like HSV-1, replicate in the nucleus and expose viral DNA in the nucleus during early infection. Paradoxically, to avoid being constitutively activated in the nucleus, cGAS is tethered to the chromatin. Our study found that HSV-1 and AdV infection caused cGAS release from the chromatin. Interestingly, RNA viruses and VACV, a DNA virus replicating in the cytosol, failed to induce nuclear soluble cGAS, suggesting the nuclear replication of DNA virus is required. Indeed, HSV-1 polymerase inhibitor blocked cGAS untethering from the chromatin. Although the mechanism of how DNA virus infection induces nuclear soluble cGAS is not clear now, our study excludes DNA damage because cisplatin could not induce nuclear soluble cGAS. In the future, it will be interesting to know whether viral proteins directly control cGAS untethering or indirectly by regulating gene expression of host factors involved in cGAS chromatin tethering.

Like many other pathogen recognition receptors, cGAS is also targeted by viral proteins to subvert host innate immune responses. For example, the UL41 and UL37 of HSV-1 induce cGAS protein degradation and deamidation to inhibit cGAS activity[40,41], respectively. In addition, HSV-1 infection also induces the acetylation of lysine 198 of cGAS, thereby impairing cGAS activation[42]. Because UL41 and UL37 are late expressing viral genes, suggesting that early detection of HSV-1 infection is critical for host innate defense. Our data provide a model that nuclear cGAS can sense HSV-1 infection at the early stage of viral infection.

A recent study showed that nuclear cGAS inhibited RNA virus infection by facilitating nuclear translocation of the protein arginine methyltransferase 5 (PRMT5)[43], which is independent of cGAS enzymatic activity. PRMT5 catalyzes the symmetric dimethylation of histone H3 at IFNβ and IFNα4 promoters, thus facilitating the expression of these IFN genes. Our data found that the enzymatic dead cGAS mutant failed to rescue nuclear soluble cGAS-mediated innate immune response to HSV-1, suggesting that cGAS enzymatic activity plays a predominant role in DNA virus infection. In addition, PRMT5 facilitates the access of IRF3 to IFN promoters, which requires the prerequisite of IRF3 phosphorylation and dimerization. Thus, our experiments might not exclude the potential role of PRMT5 since the TBK1-IRF3

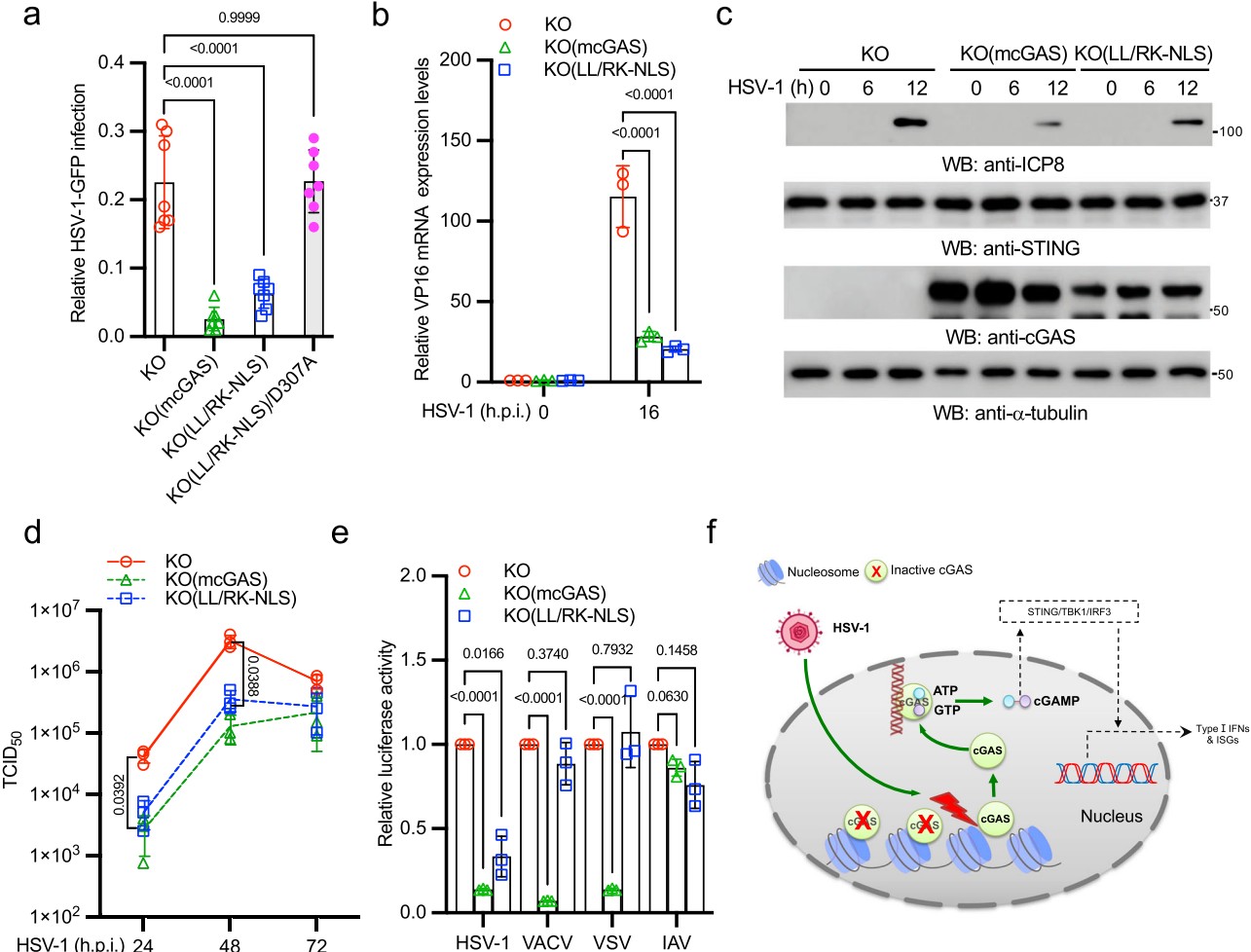

**Fig. 6 Nuclear soluble cGAS inhibits HSV-1 infection. a** cGAS KO, KO(mcGAS), KO(LL/RK-NLS), and KO(LL/RK-NLS)/D307A cells were treated with Dox for 24 h and then infected with 1 MOI of HSV-1-GFP for 16 h. Cells expressing GFP were examined and counted under a fluorescence microscope. The relative infection was determined by the ratio of positive cells. Data represent means ± s.d. of three independent experiments (> 200 cells were counted in each field and five fields were counted per experiment). The P-value was calculated by one-way ANOVA followed by Dunnett's multiple comparisons test. **b** The cGAS KO, KO(mcGAS), KO(LL/RK-NLS) RAW264.7 cells were treated with Dox for 24 h and then infected with 1 MOI of HSV-1 McKrae for 16 h. Then, cells were collected for RNA extraction. The RNA levels of HSV-1 VP16 were determined by real-time PCR. All experiments were repeated three times. The P-value was calculated by two-way ANOVA followed by Dunnett's multiple comparisons test. **c** The cGAS KO, KO(mcGAS), KO(LL/RK-NLS) RAW264.7 macrophages were treated with Dox for 24 h and then infected with 1 MOI of HSV-1 McKrae for indicated times. Cell lysates were collected and blotted as indicated. **d** The cGAS KO, KO(mcGAS), KO(LL/RK-NLS) RAW264.7 macrophages were treated with Dox for 24 h and then infected with 0.01 MOI of HSV-1 McKrae for the indicated days. HSV-1 titers were determined in Vero cells. Data represent means ± s.d. of three independent experiments. The P-value was calculated by two-way ANOVA followed by Tukey's multiple comparisons test. **e** The cGAS KO, KO(mcGAS), KO(LL/RK-NLS) RAW264.7 macrophages were treated with Dox for 24 h and then infected with HSV-1-Luc, VACV-Luc, VSV-Luc, or IAV-Gluc for 16 h. Luciferase activities were measured to determine the relative infection activity. Data represent means ± s.d. of three independent experiments. The P-value was calculated by two-way ANOVA followed by Dunnett's multiple comparisons test. **f** Model of nuclear soluble cGAS sensing DNA virus infection.

signaling axis is not activated in the enzymatic dead cGAS mutant cells. Nonetheless, it will be interesting to investigate other potential roles of nuclear cGAS in the future.

In summary, we have demonstrated that HSV-1 infection induces the release of cGAS from the chromatin to the nuclear soluble fraction and the nuclear soluble cGAS is a nuclear DNA sensor during viral infection. These findings not only uncover a biological role of nuclear cGAS but also provide insights on the design of potential therapeutics for autoimmune and infectious diseases.

## Materials and methods

**Cells**. HEK293 cells (ATCC, # CRL-1573), RAW 264.7 (ATCC, # TIB-71), NCl-H596 cells (ATCC, HTB-178), HFF-1 (ATCC, #SCRC-1041), L929 (ATCC, # CRL-6364), MDA-MB-231 (Sigma, 92020424-1VL), and Vero cells (ATCC, # CCL-81) were maintained in Dulbecco's Modified Eagle Medium (DMEM) (Life

Technologies, # 11995-065) containing antibiotics (Life Technologies, # 15140-122) and 10% fetal bovine serum (FBS) (Life Technologies, # 26140-079). NCI-H1299 cells (ATCC, # CRL-5083) and THP-1 cells (ATCC, TIB-202) were cultured in RPMI Medium 1640 (Life Technologies, # 11875-093) plus 10% FBS. A549 cells (ATCC, # CCL-185) were cultured in RPMI Medium 1640 plus 10% FBS and 1 × MEM Non-Essential Amino Acids Solution (Life Technologies, # 11140-050).

**Viruses**. HSV-1 KOS (#VR-1493) and AdV (#VR-5) were purchased from ATCC. HSV-1 KOS d109 mutant, HSV-1 Mckrae strain, HSV-1-GFP-Luc, VSV-Luc, VACV-Luc, IAV-Gluc were reported before[44,45]. Viral titration was performed as the following. Vero cells were infected with a serial diluted HSV-1. After 1 h, the medium was removed and replaced by the DMEM plus 2% FBS or 1% agarose. Cells were examined for cytopathic effects to determine TCID50 or plaque assays for viral titers.

**BMDM preparation**. Bone marrow cells were isolated from femur and tibia of 6-8 weeks old C57BL/6 J mice, and then were plated in 10-cm petri dishes and cultured in the DMEM medium containing 10% FBS and 30% L929-conditioned medium.

**Plasmids**. Mouse cGAS cDNA were synthesized and cloned into pCMV-3Tag-8 to generate mcGAS-FLAG. The point mutations (mcGAS-LL/RK, mcGAS-R241E, and mcGAS-D307A) and deletion mutants (hcGAS-N, hcGAS-delN, hcGAS-delNES, and mcGAS-delN) were constructed using a Q5® Site-Directed Mutagenesis Kit (New England Biolabs, # E0554S). The nuclear localization signal (NLS) of SV40 large T was fused to the C-terminus of mouse cGAS and its mutants to generate mcGAS-LL/RK-NLS (LL/RK-NLS) and mcGAS-LL/RK-NLS-D307A (LL/RK-NLS/D307A), and then cloned into pCW57-MCS1-P2A-MCS2(Hygro) (Addgene, #80922).

hcGAS cloning primers: hcGAS-XhoIF 5'- gcactctgaggccaccatgcagccttggcacgg aaag -3' and hcGAS-NotIR 5'- gtagcggccgcaaattcatcaaaaactgg -3'. hcGAS-N cloning primers: hcGAS-XhoIF and hcGAS-160-NotIR 5'- gtagcggccgcaggcgccgcatccctc cgtac -3'. hcGAS-delN cloning primers: hcGAS-161-XhoIF 5'- gcactctgaggccaccatg ggggcctcgaagctccgggcg 3' and hcGAS-NotIR. hcGAS-delNES mutagenesis primers: 5'- agccgcgatgatatctccacg -3' and 5'- aaccgcccggagcttcgaggc -3'.

mcGAS cloning primers: mcGAS-XhoIF 5'- gtactcgaggccaccatggaagatccg cgtagaaggac -3' and mcGAS-NotIR 5'- gtagcggccgcaagcttgtcaaaaattggaaac -3'. mcGAS-delN cloning primers: mcGAS-146-XhoF 5'- gtactcgagatggaaccgga caagctaaagaagg -3' and mcGAS-NotIR. mcGAS-NLS cloning primers: mcGAS-XhoIF and mcGAS-NotIR-NLS 5'- gtagcggccgcatccagtttcttttttcttagctgccgggct aagcttgtcaaaaattgga -3'. mcGAS R241E mutagenesis primers: 5'- gagattccacga ggaaatccgctgag -3' and 5'- tttgaatttcacaagatagaaagcacc -3'. mcGAS LL/RK mutagenesis primers: 5'- agaaagaaacgcaaagatatctcggaggc -3' and 5'- cctttgtccag caccttcttttagct -3'. mcGAS D307A mutagenesis primers: 5'- gctataattctggctttggag -3' and 5'- cacagagatttcttcagggttc -3'.

**Antibodies**. Primary antibodies: Anti-β-actin [Abcam, # ab8227, WB (1:10,00)], anti-FLAG [Sigma, # F3165, WB (1:10,00), IFA (1:100)], anti-TBK1 [Cell Signaling Technology, # 3504 S, WB (1:10,00)], anti-phospho-TBK1 (Ser172) [Cell Signaling Technology, # 5483 S, WB (1:10,00)], anti-C1QBP [Cell Signaling Technology, # 6502 S, WB (1:10,00)], anti-α-Tubulin [Cell Signaling Technology, # 3873 S, WB (1:10,00)], anti-Histone H3 [Cell Signaling Technology, # 4499 S, WB (1:10,00)], anti-STING [Cell Signaling Technology, # 50494 S, WB (1:10,00)], anti-vimentin [R&D Systems, MAB21052-SP, WB (1:10,00)], anti-ICP8 [Abcam, ab20194, WB (1:10,00)], anti-γ-H2AX [ABclonal, AP0099, WB (1:10,00)], anti-HDAC2 [Cell Signaling Technology, #5113, WB (1:10,00)], anti-human cGAS [Cell Signaling Technology, #79978, WB (1:10,00), IFA (1:100)], anti-mouse cGAS [Cell Signaling Technology, #31659, WB (1:10,00)], anti-Anti-E1A [Sigma, #05-599, WB (1:10,00)], anti-VACV [Bio-Rad, #9503-2057, WB (1:10,00)], anti-IAV-NP [GenScript, # A01506-40, WB (1:10,00)], anti-VSV [Imanis Life Sciences, #REA005, WB (1:10,00)].

Secondary antibodies: Goat anti-Mouse IgG-HRP [Bethyl Laboratories, # A90-116P, WB (1:10,000)], Goat anti-Rabbit IgG-HRP [Bethyl Laboratories, # A120-201P, WB (1:10,000)], Alexa Fluor 594 Goat Anti-Mouse IgG (H + L) [Life Technologies, # A11005, IFA (1:200)], Alexa Fluor 488 Goat Anti-Rabbit IgG (H + L) [Life Technologies, # A11034, IFA (1:200)].

**Subcellular fractionation**. H1299, HEK293, THP-1, and BMDM cells were transfected with 1 μg/mL ctDNA or infected with indicated virus for designated times. After washing with 1 x PBS, approximately $2 \times 10^6$ cells were harvested. Cells were fractionated using the Subcellular Protein Fractionation Kit for Cultured Cells (Thermo scientific, #78840). The fractions used for Western blotting were immediately mixed with the Lane Marker Reducing Sample Buffer (Thermo Fisher Scientific, # 39000) and boiled at 95 °C for 5 minutes. The fractions used for cGAMP assay were stored at −80 °C until use.

**ChIP-qPCR assay**. Briefly, HEK293 cells stably expressing FLAG-tagged mcGAS-LL/RK-NLS were infected with HSV-1-GFP virus for 16 h. Proteins were cross-linked to DNA by adding formaldehyde drop-wise directly to the media to a final concentration of 0.75% and rotate gently at room temperature (RT) for 10 min. Glycine was added to the media to a final concentration of 125 mM for 5 min at RT to stop the cross-linking. Cells were then rinsed two times with 1 mL cold PBS and harvested by scraping. Cells were transferred into 1.5 mL tube and centrifuged for 5 min at 4 °C, 1000 x g. Cells were fractionated using the Subcellular Protein Fractionation Kit for Cultured Cells. The nuclear soluble extracts were diluted 15 times with tandem affinity purification (TAP) lysis buffer [50 mM Tris-HCl (pH 7.5), 10 mM MgCl₂, 100 mM NaCl, 0.5% Nonidet P40, 10% glycerol, the Complete EDTA-free protease inhibitor cocktail tablets (Roche, # 11873580001)]. Then 10 μL anti-FLAG magnetic beads (Sigma, M8823) were added into the mixture and followed by incubating at 4 °C overnight with gently rotating. The beads were centrifuged at 3000 rpm for 1 min at RT. The supernatant was discarded. The beads were then suspended and washed three times with 1 x PBS through gently rotating at RT. Then the beads were collected through centrifuging at 3000 rpm for 1 min at RT and the cGAS complexes were eluted using 40 μL 0.2 mg/mL 3 x FLAG peptides, and then the elutes were subject to qPCR assay.

**cGAMP assay**. Cells were collected by centrifugation at 500 x g for 5 minutes, and then resuspended and lysed in PBS or the Immunoassay Buffer C (Cayman chemical, # 401703) through boiling at 95 °C for 10 minutes. The lysates were then

centrifuged for 30 min at 4 °C, 10,000 x g. Supernatants were collected for cGAMP ELISA assays. For subcellular fractions, the samples were diluted 10 times with the Immunoassay Buffer C. The cGAMP amount was determined by ELISA assays according to the manufacture's protocols (Cayman chemical). Briefly, the strips were pre-washed five times with Wash Buffer prior to be used in the ELISA. The Immunoassay Buffer C, 2'3'-cGAMP ELISA Standard, Samples, 2'3'-cGAMP-HRP Tracer and the 2'3'-cGAMP ELISA Polyclonal Antiserum reagents were added into wells accordingly. The strips were incubated overnight at 4 °C. After rinsing five times with 300 μL Wash Buffer (1x), 175 μL of the TMB Substrate Solution was added to each well. After 30 min, 75 μL of HRP Stop Solution was added to each well. Plate was read at a wavelength of 450 nm. The 2'3'-cGAMP concentration was calculated according to the manufacture's instruction.

**Sample preparation, Western blotting, and immunoprecipitation**. Approximately $1 \times 10^6$ cells were lysed in 500 μL of TAP lysis buffer for 30 min at 4 °C. The lysates were then centrifuged for 30 min at 15,000 rpm. Supernatants were mixed with the Lane Marker Reducing Sample Buffer and boiled at 95 °C for 5 minutes. Western blotting and immunoprecipitation were performed as the following procedures[46]. Briefly, samples (10–15 μL) were loaded into Mini-Protean TGX Precast Gels, 15 well (Bio-Rad, # 456-103), and run in 1 × Tris/Glycine/SDS Buffer (Bio-Rad, # 161-0732) for 60 min at 140 V. Protein samples were transferred to Immun-Blot PVDF Membranes (Bio-Rad, # 162-0177) in 1 × Tris/Glycine buffer (Bio-Rad, # 161-0734) at 70 V for 60 min. PVDF membranes were blocked in 1 × TBS buffer (Bio-Rad, # 170-6435) containing 5% Blotting-Grade Blocker (Bio-Rad, # 170-6404) for 1 h. After washing with 1 × TBS buffer for a total of 30 min (10 min each time, repeat 3 times), the membrane blot was incubated with the appropriately diluted primary antibody in antibody dilution buffer (1 × TBS, 5% BSA, 0.02% sodium azide) at 4 °C for 16 h. Then, the blot was washed three times with 1 × TBS (each time for 10 min) and incubated with secondary HRP-conjugated antibody in antibody dilution buffer (1:10,000 dilution) at RT for 1 h. After three washes with 1 × TBS (each time for 10 min), the blot was incubated with Clarity Western ECL Substrate (Bio-Rad, # 170-5060) for 1-2 min. The membrane was removed from the substrates and then exposed to the Amersham imager 600 (GE Healthcare Life Sciences, Marlborough, MA).

**Immunofluorescence assay**. Cells were cultured in the Lab-Tek II CC2 Chamber Slide System 4-well (Thermo Fisher Scientific, # 154917). After the indicated treatment, the cells were fixed and permeabilized in cold methanol for 10 min at −20 °C. Then, the slides were washed with 1 × PBS for 10 min and blocked with Odyssey Blocking Buffer (LI-COR Biosciences, # 927-40000) for 1 h. The slides were incubated in Odyssey Blocking Buffer with appropriately diluted primary antibodies at 4 °C for 16 h. After 3 washes (10 min per wash) with 1 × PBS, the cells were incubated with the corresponding Alexa Fluor conjugated secondary antibodies (Life Technologies) for 1 h at room temperature. The slides were washed three times (10 min each time) with 1 × PBS and counterstained with 300 nM DAPI for 1 min, followed by washing with 1 × PBS for 1 min. After air-drying, the slides were sealed with Gold Seal Cover Glass (Electron Microscopy Sciences, # 3223) using Fluoro-gel (Electron Microscopy Sciences, # 17985-10).

**Luciferase assays**. Cells were infected with different reporter viruses for indicated times. Then, cells were lysed, and cell lysates were collected for luciferase activity assays using the Dual Luciferase reporter system (Promega, #E1980) as instructed by the manufacturer.

**MTT assays**. Cells were seeded in a 96-well plate at a density of 400 cells/well in 100 μL culture medium. Then, cells were treated with DMF, cisplatin, or HSV-1. Cell viability was determined by the MTT Assay Kit (Cayman Chemical, #10009365).

**Real-time PCR**. Total RNA was prepared using the E.Z.N.A. Total RNA Kit I (Omega Bio-Tek Inc, R6834). 400 ng of RNA was reverse transcribed into cDNA using the ProtoScript II First Strand cDNA Synthesis Kit (New England Biolabs®, E6560L). For one real-time reaction, 5 μL of Maxima SYBR Green/ROX qPCR Master Mix (2×) (Thermo Scientific), with 100 ng of the synthesized cDNA plus an appropriate random primer mix, were analyzed on a C1000 Touch Thermal Cycler CFX96 Real-Time System (Bio-Rad). The comparative Ct method was used to determine the relative mRNA expression of genes normalized by the housekeeping gene *GAPDH*. The primer sequences: mouse *Gapdh*, forward primer 5'- gcggcacgtcagatcca -3', reverse primer 5'- catggccttccgtgttccta -3'; mouse *Ifnb1*, forward primer 5'- cagctccaagaaaggacgaac -3', reverse primer 5'- ggcagtgtaactcttctgcat -3'; mouse *Cxcl10* (IP10), forward primer 5'- ccaagtgctgccgtcattttc -3', reverse primer 5'- ggctcgcagggatgatttcaa -3'; mouse *Ccl5* (RANTES), forward primer 5'- gctgctttgcctacctctcc -3', reverse primer 5'-tcgagtga-caaacacgactgc-3'; GFP, forward primer 5'- aagggcgactcaagg -3', reverse primer 5'- tgcttgtcggccatgatatag -3'; HSV-1 VP16, forward primer 5'- ggactgtattccagcttcac -3', reverse primer 5'- cgtcctcgccgtctaagtg -3'.

**Transfection**. HEK293 cells were transfected using Lipofectamine 3000 or Lipo-fectamine LTX Transfection Reagent (Life Technologies, # L3000015) according to the manufacturer's protocol.

Calf thymus DNA (ctDNA) was purchased from R&D systems (# 9600-5-D). For cytosolic DNA stimulation, ctDNA was transfected into cells using polyethylenimine (PEI) (Polysciences, #23966).

**CRISPR/Cas9**. The mouse cGAS single guide RNA targeting sequence: 5'- GC GGACGGCTTCTTAGCGCG -3'. The sgRNA was cloned into lentiCRISPR v2 vector. The lentiviral construct was transfected with psPAX2 and pMD2G into HEK293T cells using PEI. After 72 h, the media containing lentivirus were collected. The targeted cells were infected with the media containing the lentivirus supplemented with 10 μg/mL polybrene. Cells were selected with 10 μg/mL puromycin for 14 days. Single clones were expanded for knockout confirmation by Western blotting.

**Stable cell line selection**. HEK293 cells and RAW 264.7 cells were transfected with the relative constructs or infected with the media containing the lentivirus supplemented with 10 μg/mL polybrene. Cells were selected with 200 μg/mL hygromycin or 10 μg/mL puromycin for 14 days. Stable cell lines were validated by Western blotting.

**Statistics and reproducibility**. Statistical analysis was performed as stated in the methods of figure legends on at least three independent biological replicates. Unless otherwise stated, all experiments were performed with $n \geq 3$ biological replicates.

**Reporting summary**. Further information on research design is available in the Nature Research Reporting Summary linked to this article.

## Data availability

All relevant data are within the manuscript and its supplementary files. The constructs of pCW57-mcGAS, pCW57-mcGAS-LL/RK-NLS, and pCW57-mcGAS-LL/RK-NLS/D307A are deposited in Addgene with ID numbers 185458, 185457, and 185459, respectively. Source data for figures are available in Supplementary Data 1. Uncropped of western blots are provided in Supplementary Figs. 7–11.

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

## Acknowledgements
This research was funded by the National Institutes of Health (R21AI137750, R21AI166043, R21AI167870, and R01AI141399 to S.L.).

## Author contributions
SL conceived and supervised the project. S.L., Y.W., K.S., and L.W. designed the study. Y.W., K.S., W.H., J.L., and L.W. performed the experiments. S.L., Y.W., K.S., and L.W. analyzed the data. All authors contributed to manuscript writing, revision, read, and approved the submitted version.

## Competing interests
The authors declare no competing interests.
