## [Peer Review File · Communications Biology]

Reviewers' comments:

Reviewer #1 (Remarks to the Author):

In this manuscript, Wu et al., show that infection with DNA viruses such as adenovirus and herpes simplex virus 1 (HSV-1) liberate chromatin-bound DNA sensor cyclic GMP-AMP synthase (cGAS) in the nuclear soluble fraction. The nuclear soluble cGAS, in turn, limits HSV-1 infection by producing cGAMP and inducing the expression of type I IFN and interferon-stimulated genes such as IP-10 and RANTES. The authors provide convincing evidence, supported by the use of multiple cell lines, primary mouse cells, genetic knockout system, inducible overexpression system, and pharmacological inhibitors.

Overall, based on conceptual advance and technical soundness, I find this manuscript a valuable contribution to the field of innate immunity. The following suggestions might further strengthen the conclusions of this study.

Comments to the authors:

1. The authors provide substantial evidence that in response to HSV-1 infection, nuclear soluble cGAS produces cGAMP and activates type I IFN and interferon-stimulated genes such as IP-10 and RANTES. However, cGAMP requires the stimulator of interferon genes (STING) to drive the expression of type I IFN, IP-10 and RANTES (Decout et al., Nat Rev Immunol. 2021 Sep;21(9):548-569). Given that STING was not observed in the nuclear soluble fraction upon HSV-1 infection, the authors should explain how cGAS drives the downstream signalling within the nucleus (presumably without STING?).

The contribution of STING can be addressed by using HEK293T cells. As compared to HEK293, HEK293T cells lack endogenous STING and do not produce IFNs in response to cytosolic DNA stimulation (Sui et al., Sci Signal. 2017;10(488):eaah5054). The authors could utilise the inducible cGAS expression system in HEK293T cells followed by HSV-1 infection and assessment of type I IFN, IP-10 and RANTES. Given that cGAS may function independently of STING during certain disease conditions such as colorectal cancer (Hu et al., Proc Natl Acad Sci U S A. 2021;118(23):e2105747118), determining whether the nuclear cGAS requires STING during infection could further strengthen the study and will provide valuable mechanistic insight into the role of cGAS during infection.

2. A recent study suggests that in addition to the cytosol, STING is also located in the inner nuclear membrane and translocates out of the nucleus upon dsDNA transfection (Dixon et al., iScience. 2021;24(9):103055). Is it possible that cGAMP produced by nuclear soluble cGAS binds to STING on the inner nuclear membrane, which may result in cytosolic translocation, ultimately driving downstream signalling to produce type I IFN, IP-10 and RANTES?

3. Given that an inducible cGAS expression system was established in HEK293 cells, please provide data on the production of type I IFN, IP-10 and RANTES to clarify if there were any differences between HEK293 and RAW264.7 cells.

4. The authors should revisit the statistical tests used in each panel and provide relevant details in the figure legend. Some panels do not have information about the statistical tests used. For example:

a) Figure 3c does not indicate whether data are presented as SD or SEM and which statistical test was used.

b) Figure S3c, "cytosol" panel should include statistical analysis.

c) An appropriate test for Figure S3d would be one-way ANOVA instead of two-way ANOVA.

5. The model describing the findings of the study in Figure 6f is oversimplified. The authors should consider adding type I IFN, IP-10 and RANTES to this model. The suggestions in "comment 1/2" could help in modifying this model.

6. For data reproducibility, I suggest adding information related to the source of ctDNA (cytoplasmic DNA) used in this study.
7. For enhanced readability, I suggest re-checking/revising a few sentences. For instance, lines 33, 195, 206, and 293.
8. Line 178, "...HEK cells that are lacking endogenous cGAS", should be supported by reference and/or referred to Figure S4a.
9. The legend of Figure S3 (line 640) should indicate if dimethylformamide (DMF) was used as a vehicle (mock).

Reviewer #2 (Remarks to the Author):

The Review:

The manuscript entitled " Nuclear soluble cGAS senses DNA virus infection ", from laboratory of Shitao Li, provides strong and compelling evidence that DNA virus (HSV-1) can be sensed by nuclear soluble cGAS in RAW 264.7 and HEK293 cells overexpressing cGAS. The authors also show that cGAS is found in nuclear soluble fraction in THP-1 and BMDM, however the function/activity of nuclear cGAS in these cells remain unknown. They have also shown that when the HSV-1 replication is blocked by acyclovir, hence with little or no viral DNA in the nucleus, the release of cGAS from the chromatin is inhibited. They do not have a mechanism for how the cGAS is released from the chromatin in a manipulated and HSV-1 infected cells. Finally, they show that the nuclear cGAS has an antiviral effect against HSV-1 by upregulating IFN β /IP-10 mRNA expression. The authors progress in a systematic way to explore the characteristics of nuclear cGAS and HSV-1 infection. With minor exceptions, the manuscript is clearly written and the figures, although complex, are well laid out. The authors properly discuss the results in the context of the existing literature and address limitations of their study, not overstating their findings. The novelty lies in finding a role for nuclear cGAS in sensing HSV-1. The claims are convincing, however some experiments need some improvement. The material and method section is not presented in sufficient methodological detail, so that the experiments could be reproduced (The luciferase assay is not even mentioned). Most experiments are reproduced three times, and often n=3. My major concern is that they do not provide any mechanism on how and why the cGAS is released from the nucleus. However, it suitable for the journal and would advise to accept this manuscript with revisions.

Major points:

1. When cytosolic DNA stimulation is used, the authors add ctDNA to the cytosol. Major concern: Does this stimulation work at all? What is ctDNA? Please show that ctDNA added to the supernatant (not transfected ctDNA) does get into the cytoplasm and stimulates cGAS in one of the cell lines which has cGAS in the cytoplasm. Please also show that when ctDNA is added to the RAW 264.7 cell culture media the cells do respond by either performing qPCR on an ISG (e.g. IP-10, ISG15), or simply measuring cGAMP release like in fig S3b.
2. The authors need to clarify why (and when) they chose to use McKrae, KOS, KOS-d109. The same goes for why they chose to transfect ctDNA in some case instead of adding it to the media. When do they chose to use HEK and when RAW cells? When did they use murin cGAS and when human cGAS? Reason for the choice they make.
3. When the authors use IFA, they do not add ctDNA to the supernatant, but transfect it into the cells. Is there a reason for this?
4. Fig S3C. The authors claim to measure the activity of cGAS from different fractions, but they measure cGAMP levels in different fractions. The cGAS activity is not the same as measuring cGAMP levels in different fractions, since the cGAMP produced in different compartments of the cells can move to the cytoplasm. If the cGAS activity from different compartments were to be measured, we suggest to stimulate the different fractions with DNA and then measure cGAMP

release.

5. It is confusing to understand when the mouse or when the human cGAS plasmids are used. It could be nice to have an overview of all the different cGAS constructs used. Generally, detailed information about how all the deletion and pointmutation constructs are made is lacking.

6. Fig2. It is nice to see that the authors have included different viruses. Keep in mind that not all viruses have tropism or the ability to infect. Some cells require higher MOI or different time points to activate cGAS. It could be nice to see a WB in fig 2b,c,d,e showing that these cells are actually infected. If it is difficult to get Ab for each virus, this could also be shown by qPCR on viral RNA/DNA, together with induction of an ISG (e.g. CXCL10, ISG15) from the infected cells (like in fig S5b). Or, simply measure cGAMP (like in fig S3b). -Why is AdV, which is the only other virus then HSV that can release cGAS into nuclear soluble fraction, not included in fig S3d, Fig6E?

7. In order to find out if the HSV-1 replication or the HSV DNA by itself can activate nuclear cGAS: Transfect with HSV-1 DNA fragments in the cells to evaluate if the nuclear cGAS activation is depended on the HSV-1 sequence.

8. Page 8 line 170: "To exclude the effects of apoptosis on cGAS subcellular localization, we examined cell death caused by cisplatin and HSV-1." The MTT assay is not specific for apoptosis, please reword to include other types of cell death. If apoptosis is the aim, they should measure Annexin/PI staining or cleaved caspase 3 assay.

Minor points:

Several papers have shown that Sting is degraded after HSV-1 infection (not observed in any of the WB). Several papers have also shown that cGAS is degraded or inactivated. Please comment on this in context of the finding here.

- HSV-1 UL41 protein has been shown to degrade cytosolic cGAS (doi: 10.1128/JVI.02414-16).

- HSV-1 UL37 protein has also been shown to inactivate cGAS (doi: 10.1016/j.chom.2018.07.004).

- Nuclear cGAS Functions Non-canonically to Enhance Antiviral Immunity via Recruiting Methyltransferase Prmt5. <https://doi.org/10.1016/j.celrep.2020.108490>: This paper shows that the nuclear-localized cGAS activates the innate immune response through a DNA sensing-independent mechanism in various cells under RNA virus infection.

- hcGAS-Lys198 acetylation was found to be decreased by quantitative proteomics upon infection by either HSV-1 or HCMV (human cytomegalovirus), suggesting that these DNA viruses might hijack this acetylation regulation to specifically inactivate cGAS to evade innate-immune surveillance.

- In the nucleus, cGAS interacts with replication fork proteins in a DNA binding-dependent manner. The binding of cGAS to DNA slows replication forks. DOI: 10.1126/sciadv.abb8941: Is it possible that this is occurring during viral DNA replication?

Fig 6. It is nice to see that the viral replication is repeated by several methods to validate the finding.

Fig1: Confirming previous published data.

FigS1. It could be nice to have IFA using RAW 264.7, which is predominantly used in the paper.

Fig S2: Please indicate the MOI used for infection.

Fig S2b + d: Why is Histone present in the cytoskeletal fraction BMDM and RAW cells, while it is not present in fig. S2a???

FigS2D. Nice to include the ICP8 blot also.

Fig3a. Note that HSV-1 d109 mutant is on KOS strain.

Fig3b. Which HSV-1 is used KOS/McKrea?

Fig 3c+d. In fig 3d, it is shown that around 80% of the cells are dead with cisplatin treatment after 12hr. Is it then possible to do fractionation (fig 3C) on 80% dead cells and conclude that DNA damage does not lead to cGAS release from the chromatin?

Fig.S3b: It is essential to know if the DNA is transfected or added to the media. Reaction mix?

Fig S3C: According to fig 2a, Mock Raw cells have equal amounts of cGAS in the cytoplasm as

chromatin bounds. Fig S3C shows that only cytoplasmic cGAS is active.

Fig S3D. Could be nice to have a DNA virus, which replicates in the nucleus, but not essential.

Fig 4: Mouse or human cGAS? Fig 4 could be moved to supplemental fig.

FigS4d: Lacks mock-infected control to evaluate the IFN/IGS induction after infection with HSV-1-d109.

FigS5b: It is nice to see that ctDNA added to the media does not reach into the nucleus to activate the nuclear-cGAS. It could be nice to see this with WT-RAW cells.

Fig6e. HSV-1-Luc, VACV-Luc, VSV-Luc, or IAV-Gluc - no info on the virus in the material and methods.

It could be nice to elucidate the role for nuclear cGAS in physiological conditions in cells that normally recognize HSV-1. HSV-1 is a neurotropic virus and the infection is often found in the brain. It has been shown that microglia cells in the brain are mainly responsible for CGAS dependent recognition of HSV-1 (<https://doi.org/10.1038/ncomms13348>). Therefore, it could be relevant to study the role of nuclear cGAS in microglia cells.

Reviewer #3 (Remarks to the Author):

Yakun et al. report to use the sub-cellular fractionation to identify the nuclear soluble cGAS upon HSV infection. cGAS has been characterized to freely sense dsDNA in the cytosol and stay inactive with chromatin in the nucleus. It has been well established that HSV could activate the cGAS-STING pathway, but the mechanism remains unclear in that HSV replicates in the nucleus, where free cGAS does not exist. Yakun et al. target the focused question in the field and characterize the HSV-induced nuclear soluble cGAS senses virus infection. The LL/RK-NLS cGAS prevents interference with cytosolic cGAS, making this conclusion solid. This work is well controlled and presented in general. I have only very minor comments for the authors that might guide small improvements during preparation of a final version of the manuscript. In summary, this is an impressive piece of work.

1) Is the untethering of cGAS specific to HSV/AdV infection or all other DNA viruses replicating in the nucleus? If no more experiment or mechanism is provided, the title and conclusions should be more restrictive.

Title: Nuclear soluble cGAS senses DNA virus infection

Responses to the Reviewers comments (comments from reviewers in black, responses to the reviewers in blue, text changes are highlighted in the manuscript)

We thank the reviewers for the constructive comments and suggestions to improve our manuscript. We agreed to most of the suggestions and comments. We have carried out additional experiments and made modifications in the text to improve the manuscript quality. Our point-by-point responses are included as the following.

Reviewer comments:

Reviewer #1 (Remarks to the Author):

In this manuscript, Wu et al., show that infection with DNA viruses such as adenovirus and herpes simplex virus 1 (HSV-1) liberate chromatin-bound DNA sensor cyclic GMP-AMP synthase (cGAS) in the nuclear soluble fraction. The nuclear soluble cGAS, in turn, limits HSV-1 infection by producing cGAMP and inducing the expression of type I IFN and interferon-stimulated genes such as IP-10 and RANTES. The authors provide convincing evidence, supported by the use of multiple cell lines, primary mouse cells, genetic knockout system, inducible overexpression system, and pharmacological inhibitors.

Overall, based on conceptual advance and technical soundness, I find this manuscript a valuable contribution to the field of innate immunity. The following suggestions might further strengthen the conclusions of this study.

We appreciate the reviewer for comments on our manuscript a valuable contribution to the field of innate immunity.

Comments to the authors:

1. The authors provide substantial evidence that in response to HSV-1 infection, nuclear soluble cGAS produces cGAMP and activates type I IFN and interferon-stimulated genes such as IP-10 and RANTES. However, cGAMP requires the stimulator of interferon genes (STING) to drive the expression of type I IFN, IP-10 and RANTES (Decout et al., Nat Rev Immunol. 2021 Sep;21(9):548-569). Given that STING was not observed in the nuclear soluble fraction upon HSV-1 infection, the authors should explain how cGAS drives the downstream signalling within the nucleus (presumably without STING?).

The contribution of STING can be addressed by using HEK293T cells. As compared to HEK293, HEK293T cells lack endogenous STING and do not produce IFNs in response to cytosolic DNA stimulation (Sui et al., Sci Signal. 2017;10(488):eaah5054). The authors could utilise the inducible cGAS expression system in HEK293T cells followed by HSV-1 infection and assessment of type I IFN, IP-10 and RANTES. Given that cGAS may function independently of STING during certain disease conditions such as colorectal cancer (Hu et al., Proc Natl Acad Sci U S A. 2021;118(23):e2105747118), determining whether the nuclear cGAS requires STING during infection could further strengthen the study and will provide valuable mechanistic insight into the role of cGAS during infection.

We thank the reviewer for the suggestion. We examined the HEK293 and HEK293T cells stably expressing the inducible nuclear cGAS (Fig. S5c). As shown in Figure S5e, HSV-1 KOS d109 infection induced IFN β mRNA expression in the stable HEK293 cells but not HEK293T cells, suggesting that nuclear soluble cGAS sensing is also through STING. We speculate that cGAMP might diffuse freely through nuclear pores on the nuclear membranes, which is also suggested by the Manel lab (PMID: 30811988). We add it to Discussion and modify our model accordingly in Fig. 6f.

Figure S5

2. A recent study suggests that in addition to the cytosol, STING is also located in the inner nuclear membrane and translocates out of the nucleus upon dsDNA transfection (Dixon et al., *iScience*. 2021;24(9):103055). Is it possible that cGAMP produced by nuclear soluble cGAS binds to STING on the inner nuclear membrane, which may result in cytosolic translocation, ultimately driving downstream signalling to produce type I IFN, IP-10 and RANTES?

We have noticed several papers on nuclear STING. It is plausible that nuclear cGAMP binds and activates nuclear STING. However, it is technically difficult to exclude the interference of ER-resident STING because of the continuous membrane system between the nuclear and ER membranes. To our best knowledge, there is no nuclear-only STING tool available to test this hypothesis. Future studies will investigate the role of nuclear STING.

3. Given that an inducible cGAS expression system was established in HEK293 cells, please provide data on the production of type I IFN, IP-10 and RANTES to clarify if there were any differences between HEK293 and RAW264.7 cells.

We examined IFN β mRNA expression in HE293 cells expressing the inducible LL/RK-NLS-cGAS. After doxycycline induction and HSV-1 KOS d109 infection, we found that IFN β mRNA was induced by HSV-1 infection (Fig. S5e), which is consistent with the RAW264.7 macrophages. The induction level of IFN β mRNA is much lower in HEK293 cells than in RAW264.7 cells because RAW264.7 cells are professional innate immune cells expressing much more STING as shown below.

4. The authors should revisit the statistical tests used in each panel and provide relevant details in the figure legend. Some panels do not have information about the statistical tests used. For example:

a) Figure 3c does not indicate whether data are presented as SD or SEM and which statistical test was used.

Added in Fig. 3d. The data are presented as SD.

b) Figure S3c, “cytosol” panel should include statistical analysis.

Added.

c) An appropriate test for Figure S3d would be one-way ANOVA instead of two-way ANOVA.
Corrected. We used one-way ANOVA.

5. The model describing the findings of the study in Figure 6f is oversimplified. The authors should consider adding type I IFN, IP-10 and RANTES to this model. The suggestions in "comment 1/2" could help in modifying this model.
We modified the model.

6. For data reproducibility, I suggest adding information related to the source of ctDNA (cytoplasmic DNA) used in this study.
Added in the section of Methods. The ctDNA (calf thymus DNA) we used in the experiments is Calf Thymus DNA, # 9600-5-D, from Bio-technie, R&D systems.

7. For enhanced readability, I suggest re-checking/revising a few sentences. For instance, lines 33, 195, 206, and 293.
We polished these sentences.

8. Line 178, "...HEK cells that are lacking endogenous cGAS", should be supported by reference and/or referred to Figure S4a.
The figure reference is added.

9. The legend of Figure S3 (line 640) should indicate if dimethylformamide (DMF) was used as a vehicle (mock).
DMF was used for the mock control. Added into the legend of Fig. 3c.

Reviewer #2 (Remarks to the Author):

The Review:

The manuscript entitled " Nuclear soluble cGAS senses DNA virus infection ", from laboratory of Shitao Li, provides strong and compelling evidence that DNA virus (HSV-1) can be sensed by nuclear soluble cGAS in RAW 264.7 and HEK293 cells overexpressing cGAS. The authors also show that cGAS is found in nuclear soluble fraction in THP-1 and BMDM, however the function/activity of nuclear cGAS in these cells remain unknown. They have also shown that when the HSV-1 replication is blocked by acyclovir, hence with little or no viral DNA in the nucleus, the release of cGAS from the chromatin is inhibited. They do not have a mechanism for how the cGAS is released from the chromatin in a manipulated and HSV-1 infected cells. Finally, they show that the nuclear cGAS has an antiviral effect against HSV-1 by upregulating IFN β /IP-10 mRNA expression.

The authors progress in a systematic way to explore the characteristics of nuclear cGAS and HSV-1 infection. With minor exceptions, the manuscript is clearly written and the figures, although complex, are well laid out. The authors properly discuss the results in the context of the existing literature and address limitations of their study, not overstating their findings. The novelty lies in finding a role for nuclear cGAS in sensing HSV-1. The claims are convincing, however some experiments need some improvement. The material and method section is not presented in sufficient methodological detail, so that the experiments could be reproduced (The luciferase assay is not even mentioned). Most experiments are reproduced three times, and often n=3. My major concern is that they do not provide any mechanism on how and why the cGAS is released from the nucleus. However, it suitable for the journal and would advise to accept this manuscript with revisions.

We thank the reviewer for the comments that we provide strong and compelling evidence and advising to accept our manuscript with revisions.

We have updated the Materials and Methods section.

Major points:

1. When cytosolic DNA stimulation is used, the authors add ctDNA to the cytosol. Major concern: Does this stimulation work at all? What is ctDNA? Please show that ctDNA added to the supernatant (not transfected

ctDNA) does get into the cytoplasm and stimulates cGAS in one of the cell lines which has cGAS in the cytoplasm. Please also show that when ctDNA is added to the RAW 264.7 cell culture media the cells do respond by either performing qPCR on an ISG (e.g. IP-10, ISG15), or simply measuring cGAMP release like in fig S3b.

The ctDNA used in the experiments is calf thymus DNA (ctDNA). ctDNA was transfected into cells using the transfection reagent PEI for all ctDNA stimulation experiments, as described in our previous papers (Li et. al, Immunity 2011, PMID: 21903422; Song et. al., Journal of Immunology, 2021, PMID: 34526378). We now add details in the Methods and clarifications in figure legends.

2. The authors need to clarify why (and when) they chose to use McKrae, KOS, KOS-d109. The same goes for why they chose to transfect ctDNA in some case instead of adding it to the media. When do they chose to use HEK and when RAW cells? When did they use murin cGAS and when human cGAS? Reason for the choice they make.

The ctDNA was only transfected into cells (see explanation above).

We chose McKrae and KOS strains to demonstrate that the phenomena we observed is not strain-specific, which will increase the rigor of our research. The choice of KOS d109 is because this strain induces stronger host innate immune responses. As the reviewer knows that HSV-1 subverts host innate immunity (the references mentioned in Minor Concerns), we used a mutant HSV-1 virus to get a better innate immune response. We added an explanation in the text.

We used HEK293 cells for cGAS subcellular localization due to its high transfection efficiency. But HEK293 cells are not innate immune cells, so we chose RAW cells for innate immune response and viral infection experiments. Most cGAS expression constructs are murine cGAS. Human cGAS and its mutants are only used in Figs. S1h and S1i in the human cell line HEK293 cells. We now clarify them by mcGAS (mouse cGAS) and hcGAS (human cGAS) in the figures.

3. When the authors use IFA, they do not add ctDNA to the supernatant, but transfect it into the cells. Is there a reason for this?

The ctDNA was only transfected into cells (see explanation above).

4. Fig S3C. The authors claim to measure the activity of cGAS from different fractions, but they measure cGAMP levels in different fractions. The cGAS activity is not the same as measuring cGAMP levels in different fractions, since the cGAMP produced in different compartments of the cells can move to the cytoplasm. If the cGAS activity from different compartments were to be measured, we suggest to stimulate the different fractions with DNA and then measure cGAMP release.

We performed new experiments as suggested by the reviewer. In vitro cGAS enzymatic assays found that a significantly higher level of cGAMP was generated by cGAS in the nuclear soluble extract of cells infected with HSV-1.

5. It is confusing to understand when the mouse or when the human cGAS plasmids are used. It could be nice to have an overview of all the different cGAS constructs used. Generally, detailed information about how all the deletion and pointmutation constructs are made is lacking.

Human cGAS and its mutants were only used in Figs. S1h and S1i. We now clarify it in the figures and figure legends. All primers information and details were added into the Methods.

6. Fig2. It is nice to see that the authors have included different viruses. Keep in mind that not all viruses have tropism or the ability to infect. Some cells require higher MOI or different time points to activate cGAS. It could be nice to see a WB in fig 2b,c,d,e showing that these cells are actually infected. If it is difficult to get Ab for each virus, this could also be shown by qPCR on viral RNA/DNA, together with induction of an ISG (e.g. CXCL10, ISG15) from the infected cells (like in fig S5b). Or, simply measure cGAMP (like in fig S3b). -Why is AdV, which is the only other virus then HSV that can release cGAS into nuclear soluble fraction, not included in fig S3d, Fig6E?

We have added WB data to show the infection efficiency of each virus (Figs. 2b-2e). Our manuscript focuses on HSV-1. We will test other nuclear replication DNA viruses, including AdV, in the future.

Figure 2

7. In order to find out if the HSV-1 replication or the HSV DNA by itself can activate nuclear cGAS: Transfect with HSV-1 DNA fragments in the cells to evaluate if the nuclear cGAS activation is depended on the HSV-1 sequence.

It is well established that cGAS activation is DNA sequence independent. We have shown that transfection of DNA cannot cause the “free” nuclear soluble cGAS from the chromatin, so DNA transfection cannot activate nuclear cGAS.

8. Page 8 line 170: “To exclude the effects of apoptosis on cGAS subcellular localization, we examined cell death caused by cisplatin and HSV-1.” The MTT assay is not specific for apoptosis, please reword to include other types of cell death. If apoptosis is the aim, they should measure Annexin/PI staining or cleaved caspase 3 assay.

We changed the word from apoptosis to “cell viability”.

Minor points:

Sveral papers have shown that Sting is degraded after HSV-1 infection (not observed in any of the WB). Several papers have also shown that cGAS is degraded or inactivated. Please comment on this in context of the finding here.

- HSV-1 UL41 protein has been shown to degrade cytosolic cGAS (doi: 10.1128/JVI.02414-16).
- HSV-1 UL37 protein has also been shown to inactivate cGAS (doi: 10.1016/j.chom.2018.07.004).
- Nuclear cGAS Functions Non-canonically to Enhance Antiviral Immunity via Recruiting Methyltransferase Prmt5. <https://doi.org/10.1016/j.celrep.2020.108490>: This paper shows that the nuclear-localized cGAS activates the innate immune response through a DNA sensing-independent mechanism in various cells under RNA virus infection.
- hcGAS-Lys198 acetylation was found to be decreased by quantitative proteomics upon infection by either HSV-1 or HCMV (human cytomegalovirus), suggesting that these DNA viruses might hijack this acetylation regulation to specifically inactivate cGAS to evade innate-immune surveillance.

All viruses have their own evasion strategies/antagonisms to subvert host innate immunity. Although it is not the topic we study in this manuscript, we now discuss these references in the Discussion for better understanding the relationship between HSV-1 and the cGAS sensing pathway.

The potential role of PRMT5 in nuclear cGAS antiviral activity is also discussed.

- In the nucleus, cGAS interacts with replication fork proteins in a DNA binding-dependent manner. The binding of cGAS to DNA slows replication forks. DOI: 10.1126/sciadv.abb8941: Is it possible that this is occurring during viral DNA replication?

cGAS might regulate DNA replication; however, keep in mind that the development of cGAS knockout mice, including size and growth rate, is no different from wild type mice. Many studies have shown the antiviral activity of cGAS is dependent on STING. Thus, the role of cGAS on viral DNA replication might only have a very minor impact on host defense. We added the reference in the Discussion.

Fig 6. It is nice to see that the viral replication is repeated by sveral methods to validate the finding.

Fig1: Confirming previous published data.

It is necessary to clarify/validate previous data because there are too many inconsistent conclusions on cGAS subcellular localization.

FigS1. It could be nice to have IFA using RAW 264.7, which is predominantly used in the paper.

We have tested most commercially available cGAS antibodies for IFA. Only one antibody (Cell Signaling Technology, #79978S) passed our in-house validation by using cGAS knockout cells. Unfortunately, the antibody we used is a human-specific anti-cGAS antibody.

Fig S2: Please indicate the MOI used for infection.

Added.

Fig S2b + d: Why is Histone present in the cytoskeletal fraction BMDM and RAW cells, while it is not present in fig. S2a???

Sometimes there is a spillover to the cytoskeletal fraction due to the high salt concentration buffer. We repeated the experiments and added the new ones (Figs. S2b and S2d).

FigS2D. Nice to include the ICP8 blot also.

Added.

Figure S2

Fig3a. Note that HSV-1 d109 mutant is on KOS strain.

Added.

Fig3b. Which HSV-1 is used KOS/McKrea?

Clarified.

Fig 3c+d. In fig 3d, it is shown that around 80% of the cells are dead with cisplatin treatment after 12hr. Is it then possible to do fractionation (fig 3C) on 80% dead cells and conclude that DNA damage does not lead to cGAS release from the chromatin?

The cisplatin treatment time for the fractionation experiment is 4 h when there are no observed dead cells. Details were added in the figure legend.

Fig.S3b: It is essential to know if the DNA is transfected or added to the media. Reaction mix?

Clarified. DNA was added into the *in vitro* cGAS enzymatic assay mixture. It is a confirmatory experiment for the published paper (ref. 15, PMID: 31808743).

Fig S3C: According to fig 2a, Mock Raw cells have equal amounts of cGAS in the cytoplasm as chromatin bounds. Fig S3C shows that only cytoplasmic cGAS is active.

Previous data from other groups have shown that the activity of the chromatin bounded cGAS was inhibited by nucleosome (ref. 21-28).

Fig S3D. Could be nice to have a DNA virus, which replicates in the nucleus, but not essential.

Fig 4: Mouse or human cGAS? Fig 4 could be moved to supplemental fig.

It is the mouse cGAS. Details were added in the legend. We prefer to keeping it in main figure to make it easier for readers to understand the construct because most of them might not go to supplemental figures.

FigS4d: Lacks mock-infected control to evaluate the IFN/IGS induction after infection with HSV-1-d109. Added.

FigS5b: It is nice to see that ctDNA added to the media does not reach into the nucleus to activate the nuclear-cGAS. It could be nice to see this with WT-RAW cells.

Our data showed that ctDNA cannot induce nuclear soluble cGAS (Fig. 1b). Thus, even ctDNA is in the nucleus, it cannot be accessed by nuclear cGAS because all nuclear cGAS proteins are confined in the chromatin.

Fig6e. HSV-1-Luc, VACV-Luc, VSV-Luc, or IAV-Gluc - no info on the virus in the material and methods. It could be nice to elucidate the role for nuclear cGAS in physiological conditions in cells that normally recognize HSV-1. HSV-1 is a neurotropic virus and the infection is often found in the brain. It has been

shown that microglia cells in the brain are mainly responsible for cGAS dependent recognition of HSV-1 (<https://doi.org/10.1038/ncomms13348>). Therefore, it could be relevant to study the role of nuclear cGAS in microglia cells.

The references of these viruses were added in the Material and Methods. Further study will investigate the role of cGAS in HSV-1-infected microglia cells.

Reviewer #3 (Remarks to the Author):

Yakun et al. report to use the sub-cellular fractionation to identify the nuclear soluble cGAS upon HSV infection. cGAS has been characterized to freely sense dsDNA in the cytosol and stay inactive with chromatin in the nucleus. It has been well established that HSV could activate the cGAS-STING pathway, but the mechanism remains unclear in that HSV replicates in the nucleus, where free cGAS does not exist. Yakun et al. target the focused question in the field and characterize the HSV-induced nuclear soluble cGAS senses virus infection. The LL/RK-NLS cGAS prevents interference with cytosolic cGAS, making this conclusion solid. This work is well controlled and presented in general. I have only very minor comments for the authors that might guide small improvements during preparation of a final version of the manuscript. In summary, this is an impressive piece of work.

1) Is the untethering of cGAS specific to HSV/AdV infection or all other DNA viruses replicating in the nucleus? If no more experiment or mechanism is provided, the title and conclusions should be more restrictive.

We changed the title from “Nuclear soluble cGAS senses DNA virus infection” to “Nuclear soluble cGAS senses double-stranded DNA virus infection”.

REVIEWERS' COMMENTS:

Reviewer #1 (Remarks to the Author):

The authors have carried out additional experiments and have addressed all my comments. I have no further comments on the revised version.

Reviewer #2 (Remarks to the Author):

The authors have diligently responded to reviewer critiques and substantially improved the manuscript. Even though they do not provide any mechanism on how and why the cGAS is released from the nucleus, the manuscript is recommended to be accepted.

Communications Biology

COMMSBIO-22-0060

Title: Nuclear soluble cGAS senses DNA virus infection

We are glad to hear that our manuscript will be accepted. We would like to thank the reviewers and editors again for the inputs and efforts.

Responses to the Reviewers comments

Reviewer comments:

Reviewer #1 (Remarks to the Author):

The authors have carried out additional experiments and have addressed all my comments. I have no further comments on the revised version.

Reviewer #2 (Remarks to the Author):

The authors have diligently responded to reviewer critiques and substantially improved the manuscript. Even though they do not provide any mechanism on how and why the cGAS is released from the nucleus, the manuscript is recommended to be accepted.